



# Ion's ring current: regularities of the energy density distributions on the main phase of geomagnetic storms

**Alexander S. Kovtyukh**

Skobeltsyn Institute of Nuclear Physics, Moscow State University, Moscow, 119234, Russia; kovtyukhas@mail.ru

Correspondence: Alexander S. Kovtyukh (kovtyukhas@mail.ru)

**Abstract**. Based on the results of measurements near the equatorial plane a fluxes of $H^+$ and $O^+$ ions of the ring current (RC) from the Explorer 45, AMPTE/CCE, and Van Allen Probes (A and B) satellites, a systematic analysis of spatial distributions of the energy density for these ions on the main phase of magnetic storms was carried out. The radial profile of the RC ions energy density is characterized by the maximum ($L_m$) and by the ratio of the energy densities of the ions and the magnetic field at this maximum ($\beta_m$), and at $L > L_m$ this profile is approximated by the function $w(L) = w_0 \exp(-L/L_0)$. Quantitative dependences of the parameters $L_m$, $\beta_m$, $w_0$ and $L_0$ on the $D_{st}$ index, ion energy ($E$), and magnetic local time (MLT) are obtained; these dependences are different for $H^+$ and $O^+$ ions, as well as for ions of low ($E < 60$ keV) and higher energies. A strong azimuthal asymmetry of the RC ions with $E \sim 1$–300 keV at $L > L_m$ was revealed: for $H^+ + O^+$ and $O^+$ ions, $L_0$ increases systematically with the increasing MLT from evening to midnight sector, while for $H^+$ ions $L_0$ decreases; energy density of $O^+$ ions is more uniformly distributed over MLT compared with $H^+$ ions. For $O^+$ ions with $E \sim 1$–300 keV, $\beta_m \propto L_m^{-6}$; this result shows that a deeper penetration of hot plasma into a geomagnetic trap, during strong storms, requires not only a stronger electric field of convection, but also a significant preliminary accumulation and acceleration of ions (especially $O^+$ ions) in the source of the RC. It is shown that the greater $|D_{st}|$ at the end of the main phase of storms, the smaller the contribution of ions with $E < 60$ keV and the greater the contribution of higher-energy ions to the RC energy density (the average energy of ions increases); such effect can be associated with increases of the radial diffusion of ions with the increasing the strength of storm and the main phase duration.

**Keywords**. Ring current; magnetic storms; magnetosphere.

## 1 Introduction

According to the Dessler-Parker-Skopke theorem, the magnetic effect of a symmetric ring current (RC) is determined by the total kinetic energy of its particles, which is related to the current value of the $D_{st}$ index by a simple linear relationship.

However, during the main phase of geomagnetic storms, the distributions of the RC particles are characterized by a large asymmetry in magnetic local time (MLT). In addition, during these periods, a significant contribution to the value of $D_{st}$, comparable with the magnetic effect of the RC, can be made by other current systems of the magnetosphere, first of all, the currents of the magnetotail and currents on the front of the magnetopause (the Chapman-Ferraro currents).

It should be taken into account also that even in the idealized case of a symmetrical RC and the absence other current systems, the basic DPS relation leaves many different possibility for the ionic composition of the RC and the parameters of the spatial-energy distributions of particles (for a given value of $D_{st}$).

On the data from OGO 3 (Orbiting Geophysical Observatory 3), Explorer 45, AMPTE/CCE (Active Magnetospheric Particle Tracer Explorers/Charge Composition Explorer), CRRES (Combined Release and Radiation Effects Satellite), Polar, Van Allen Probes, and other satellites, it was established that during geomagnetic storms protons ($H^+$) and oxygen ions $O^+$ with kinetic



energy $E$ from several kiloelektronvolts (keV) to 200–300 keV make the main contribution (~ 70–
80%) to the total energy of the RC ions.
On the main phase of storms, the distributions of the RC ions, as well as $D_{st}$ index, strongly
depends on variations in the parameters of the solar wind and the interplanetary magnetic field
(IMF), the state of the upper ionosphere and the plasma sheet of the magnetotail, as well as from
substorms and phase of solar activity. Spatial-energy distributions of the RC ions, mechanisms of
their formation and dynamics during storms, as well as their mathematical models are considered
in many reviews (see, e.g., Williams, 1981, 1985; Gloeckler and Hamilton, 1987; Daglis et al.,
1999; Daglis, 2001, 2006; Kovtyukh, 2001; Ebihara and Ejiri, 2003; Keika et al., 2013;
Ganushkina et al., 2015).
To construct realistic models of the RC that describes its structure during magnetic storms of
different intensity, quantitative patterns of the variations of the main parameters of the radial
profile of the RC energy density are necessary. Such patterns are of high importance both for
theoretical studies and mathematical modeling of the dynamics of the magnetosphere, and for
many applied problems.
The parameters of the ion RC are differ considerably in different storms, and these scatter are
increases with the increase of $\left|D_{st}\right|$; the scatter of these parameters substantially reduces when
separately analyzing the data obtained on the main and recovery phases of storms (Kovtyukh,
2010). This means that on the main and recovery phases of storms, the RC parameters have
fundamentally different dependences on $D_{st}$.
From the results of measurements of particle fluxes on satellites since 1965, radial profiles of
the energy density of the RC ions during storms have been constructed. According to these results,
the stronger a storm, the more intensity of the RC and the closer it approaches to the Earth (in
average). At the same time, the energy density distributions of the RC ions over drift shells are
different in different sectors of MLT, and also depend on the mass and charge of the ions, from the
energy and pitch angle ranges of the ions. During the main phase of storms, these distributions
vary rapidly.
Here we consider the quantitative regularities in the variations of the main parameters of the
radial profiles of the energy density of the RC ions on the main phase of magnetic storms. During
these periods, the conditions in the magnetosphere are very diverse, and it is very not easy to
distinguish such regularities. However, this can be done if we introduce some physical restrictions
on the geomagnetic latitude and MLT, and separate the RC ions by mass and energy.
In the following sections, the methodology of our analysis is considered and the selection of the
experimental data is carried out (Sect. 2); the regularities in variations of the main parameters of
the ion RC on the main phase of storms are determined (Sect. 3); the physical mechanisms, which
can be used to explain the patterns obtained here, are considered (Sect. 4). The main conclusions of
this work are given in Sect. 5.

## 2 Classification and selection of the experimental data

To analyze the spatial distributions of the energy density of RC ions, reliable experimental results
were used here, which were obtained near the equatorial plane in the night hemisphere of the
magnetosphere (exceptions were made for only two storms and are given for comparison with
other results). These results belong to wide ranges of $L$ shells and ion energies.
Here we consider the results of measurements for the RC ions on the main phase of storms from
the satellites Explorer 45 (Smith and Hoffman,a 1973; Fritz et al., 1974), AMPTE/CCE
(Stüdemann et al., 1986; Hamilton et al., 1988; Greenspan and Hamilton, 2000, 2002), and Van
Allen Probes (Kistler et al., 2016; Menz et al., 2017, 2019a, 2019b; Keika et al., 2018; Yue et al.,
2018, 2019). These results were obtained during eleven magnetic storms with max$|D_{st}|$ from 64 to
307 nT. These results are listed in Table 1.
The values of UT, MLT, and $|D_{st}|$ in Table 1 correspond to the times when the satellite crosses
the maximum the energy density of ion RC (drift shell $L_m$) on the main phase of the corresponding





storm. The last column of this table contains also references to the papers, from which these values
of UT, MLT, and $L_m$ were obtained.
From the results presented in the works reviewed here, the moment of the satellite crossing of
the RC maximum in some cases can be bind to UT with an accuracy of several minutes; in other
cases, this moment is determined within ~ 10 minutes.
Almost all results considered here refer to periods of the solar activity maximum (except lines
3–5, and 17 in Table 1).

**Table 1**

| | Satellites | $E$, keV | UT | MLT | max$|D_{st}|$, nT | $|D_{st}|$, nT | $L_m$ |
|---|---|---|---|---|---|---|---|
| 1 | Explorer-45 | 1–138 | 21.30 UT Dec 17, 1971 | 23.10 | 171 | 167 | 3.1 (Smith and Hoffman, 1973) |
| 2 | Explorer-45 | 1–138 | 14.00 UT Feb 24, 1972 | 22 | 86 | 83 | 3.5 (Fritz et al., 1974) |
| 3 | AMPTE/CCE | 5–315 | 15.10 UT Sept 04, 1984 | 10.30 | 64 | 46 | 4.1 (Stüdemann et al., 1986) |
| 4 | AMPTE/CCE | 1–300 | 05.00 UT Sept 05, 1984 | 17.40 | 125 | 78 | 3.4 (Greenspan and Hamilton, 2002) |
| 5 | AMPTE/CCE | 30–310 | 00.20 UT Feb 09, 1986 | 17.30 | 307 | 273 | 2.8 (Hamilton et al., 1988) |
| 6 | AMPTE/CCE | 1–300 | 10.00 UT Nov 30, 1988 | 03 | 111 | 37 | 3.4 (Greenspan and Hamilton, 2000) |
| 7 | Van Allen Probes B | 10–60 | 09.56 UT Mar 17, 2013 | 19.20 | 132 | 66 | 3.2 (Menz et al., 2017) |
| 8 | Van Allen Probes B | 10–570 | 10.09 UT Mar 17, 2013 | 20 | 132 | 70 | 3.6 (Menz et al., 2017) |
| 9 | Van Allen Probes B | 10–60 | 18.58 UT Mar 17, 2013 | 19.30 | 132 | 98 | 3.1 (Menz et al., 2017) |
| 10 | Van Allen Probes B | 10–570 | 19.00 UT Mar 17, 2013 | 19 | 132 | 98 | 3.1 (Menz et al., 2017) |
| 11 | Van Allen Probes A | 10–60 | 20.08 UT Mar 17, 2013 | 19.30 | 132 | 117 | 3.0 (Menz et al., 2017) |
| 12 | Van Allen Probes B | 1–300 | 07.45 UT June 1, 2013 | 01.20 | 124 | 122 | 3.0 (Kistler et al., 2016) |
| 13 | Van Allen Probes B | 10–600 | 16.30 UT Aug 27, 2014 | 03 | 75 | 72 | 3.6 (Yue et al., 2018) |
| 14 | Van Allen Probes B | 50–200 | 19.30 UT Mar 17, 2015 | 02 | 234 | 166 | 3.3 (Keika et al., 2018) |
| 15 | Van Allen Probes B | 50–200 | 21.30 UT Mar 17, 2015 | 18 | 234 | 190 | 3.2 (Keika et al., 2018) |
| 16 | Van Allen Probes A | 1–60 | 23.10 UT Mar 17, 2015 | 03 | 234 | 233 | 2.7 (Menz et al., 2019a,b) |
| 17 | Van Allen Probes A | 10–600 | 22.10 UT Mar 6, 2016 | 05 | 99 | 98 | 3.0 (Yue et al., 2019) |

In many works on the RC dynamics during storms, the $D_{st}^*$ index proposed in (Burton et al.,
1975) is used, in which the magnetic field of currents on the magnetopause is excluded from the
$D_{st}$. At the beginning of the main phase of storms, these currents can make a significant
contribution to the $D_{st}$ values (see, e.g., Liemohn et al., 2001). However, to the end of the storm's
main phase (for most of the RC data considered here), the contribution of these currents to the $D_{st}$
value decreases significantly (see, e.g., McPherron and O'Brien, 2001; Siscoe et al., 2002, 2005;
Kistler et al., 2016; Keika et al., 2018). Most of the experimental results considered here refer to



the end of the main phase of storms (black points in Figs. 1-6), and all the main quantitative
regularities of the space-energy structure of the RC were obtained here by these points.Therefore,
$D_{st}$ index is used here (wdc.kugi.kyoto-u.ac.jp/dst_final/index.html).

114        In all rows of Table 1, except for row 12, parameter $L_m$ of the RC is tied to the drift shells of
particles $L$ (McIlwain, 1961), and in row 12 of this table, parameter $L_m$ is tied to $L^*$ (Roederer,
1970). Near the equatorial plane at $L < 3.5$, the difference between these parameters of drift shells
is $L–L^* < 0.1$ (see Figs. 2 and 4 in Roederer and Lejosne, 2018).

## 3 Analysis of the experimental results

### 3.1 Localization of the maximum energy density of the ring current ions

In most storms, the radial energy density profile of the RC ions has one distinct maximum.
However sometimes, during the main phase of storms, several local maxima close in position and
amplitude are formed; these maxima can merge and form a plateau. In such cases, the values of $L_m$
given in Table 1 refer to the local maximum of the RC, which is the most distant from the Earth, or
to the upper boundary of the plateau (rows 8–10, and 16 in Table 1).

125        In the experiments on the Explorer 45 satellite, instruments did not allow the separation of H$^+$
and O$^+$ ions in the RC. Such separation of ions was carried out in experiments on the
AMPTE/CCE, CRRES, Polar, and Van Allen Probes satellites; it was established that at the end of
the main phase of storms, O$^+$ ions made a significant compare with protons or even the main
contribution to the RC energy density.

130        At the end of the main phase of storms, the radial energy density profiles of the H$^+$ and O$^+$ ions
of RC, for the same energy intervals, are usually close to each other in shape and their maxima
($L_m$) practically coincide with each other (see, e.g., Krimigis et al., 1985; Gloeckler et al., 1985;
Stüdemann et al., 1986; Hamilton et al., 1988; Greenspan and Hamilton, 2002; Kistler et al., 2016;
Menz et al., 2017, 2019a, 2019b; Keika et al., 2018; Yue et al., 2018, 2019). This was the case in
all the storms considered here, in the availability of simultaneous data on H$^+$ and O$^+$ ions (rows 3–
17 in Table 1).

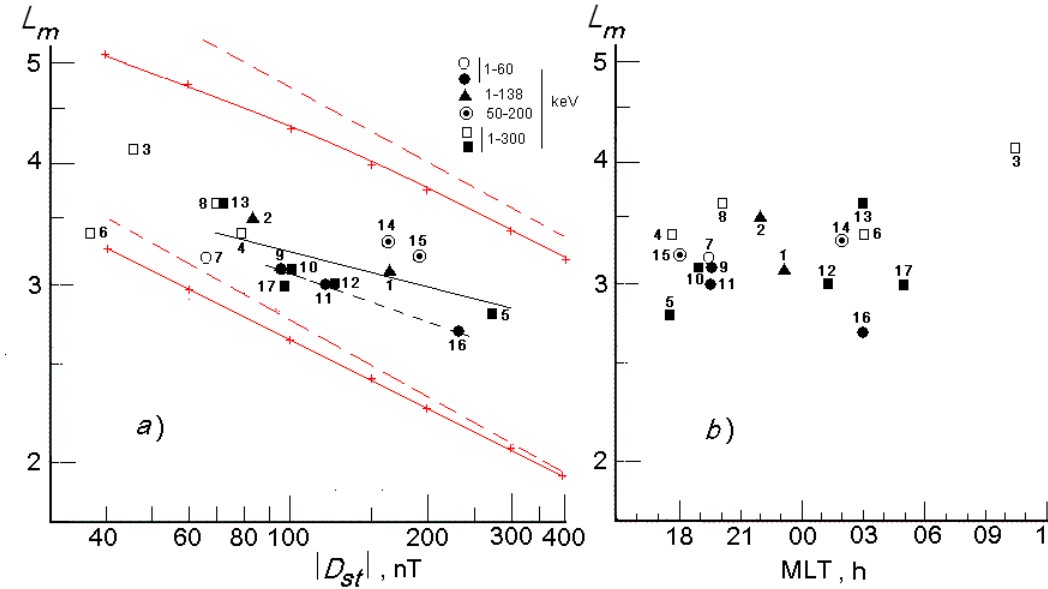


**Figure 1.** Position of the RC ions energy density maximum ($L_m$) on the main phase of various storms as the functions
of $|D_{st}|$ (*a*) and MLT (*b*).



The experimental values of parameter $L_m$ of the RC ions on the main phase of various storms
are plotted as a function of the current value of $|D_{st}|$ in Fig. 1a and from MLT in Fig. 1b. Different
symbols in Fig. 1 correspond to measurements in different ion energy ranges: ~ 1–60 keV (circles),
~ 1–140 keV (triangles), 50–200 keV (circles with a dark core), and ~ 1–300 keV (squares). Light
and dark symbols belong respectively to the middle and to the end of the main phase of storms.
The symbol numbers corresponds to the line numbers in Table 1. Such designations are carried out
in all figures of this work.
Figure 1a evidenced that with an increase in $|D_{st}|$, the average value of $L_m$ decreases. For ions
with $E \sim$ 1–60 keV, the values of $L_m$ reach their minimum values, and as the ion energy increases,
$L_m$ increases also.
Figure 1b evidenced that $L_m$ depends on MLT much weaker than on $|D_{st}|$ and ion energy. For
ions with $E \sim$ 1–300 keV, parameter $L_m(\text{MLT}) \approx$ const in the evening and near midnight sectors
(from 18 to 03 MLT).
For ions with $E \sim$ 1–300 keV in the night time MLT at the end of the main phase of storms (the
points 1, 2, 5, 10, 12, 13, 14, 15 and 17), we obtain the following approximation by least squares
method (thin black line in Fig. 1a):
$$L_m = 5.59\ |D_{st}|^{-0.117}$$

with correlation coefficient $R = -\ 0.645$. Here $D_{st}$ is in nT.
For ions with $E \sim$ 1–60 keV in the night time MLT at the end of the main phase of storms (the
points 9, 11 and 16), we obtain the following approximation (dashed black line in Fig. 1a):
$$L_m = 6.35\ |D_{st}|^{-0.157}$$

with correlation coefficient $R = -\ 0.984$. Here $D_{st}$ is in nT.
The red lines in Fig. 1a represent model dependences of parameter $L_m$ on $|D_{st}|$ (see Sect. 4.1).
**3.2 Ratio of the energy densities for ions and magnetic field at the maximum of**
**the ring current**
We use the experimental values of the ion energy density at the maximum of the RC near the
equatorial plane ($w_m$), which presented in the papers indicated in Table 1. These values are
converted to a uniform dimension (nPa).
The values of the energy density of the dipole magnetic field $w_{Bd}$ at $L = L_m$ were calculated by
the formula $w_{Bd} = bL_m^{-6}$, where $b = 3.85 \cdot 10^5$ nPa. Then the corresponding ratios $\beta_{md} = w_m/w_{Bd}$ were
calculated on $L = L_m$.
From the satellites data, during storms the magnetic field at the maximum of the RC is reduced,
and the value of this weakening is ~ 1.5 times larger compared with the $D_{st}$ values (see, e.g., Cahill
and Lee, 1975; Krimigis et al., 1985). Therefore, for the magnetic field at the RC maximum, we
consider also the values of $w_B(L_m)$ calculated by the following formula:
$$w_B(L_m) = 3.98 \cdot 10^{-4}\ (3.11 \cdot 10^4\ L_m^{-3} - 1.5\ |D_{st}|)^2.$$

Then the corresponding ratios $\beta_m = w_m/w_B$ were calculated on $L = L_m$.
The results of calculations of the parameters $\beta_{md}$ and $\beta_m$ at the RC maximum are presented in
Table 2. This table presents also the experimental values of $L_m$ (according to Table 1) and the
values of $w_m$ at the maximum of the RC (without separation by ion mass in rows 1, 2, 6, and 16,
and as the sums of the terms for $H^+$ and $O^+$ ions in other rows). The fourth and fifth columns of this
table present the calculated values of $\beta_{md}$ and $\beta_m$ at $L = L_m$ for the sum of $H^+$ and $O^+$ ions, and the
sixth and seventh columns present the values of $\beta_m$ separately for $H^+$ and $O^+$ ions.





**Table 2**

|  | $L_m$ | $w_m$, nPa | $\beta_{md}$ (H$^+$+O$^+$) | $\beta_m$ (H$^+$+O$^+$) | $\beta_m$ (H$^+$) | $\beta_m$ (O$^+$) |
|---|---|---|---|---|---|---|
| 1 | 3.1 | 50 | – | – | 0.188 | – |
| 2 | 3.5 | 20 | – | – | 0.136 | – |
| 3 | 4.1 | 5.5+5.5=11 | 0.136 | 0.189 | 0.0945 | 0.0945 |
| 4 | 3.4 | 34+10=44 | 0.177 | 0.260 | 0.201 | 0.059 |
| 5 | 2.8 | 80+160=240 | 0.300 | 0.594 | 0.198 | 0.396 |
| 6 | 3.4 | 48.3 | 0.194 | 0.224 | – | – |
| 7 | 3.2 | 10+22=32 | 0.089 | 0.111 | 0.035 | 0.076 |
| 8 | 3.6 | 9+24=33 | 0.187 | 0.263 | 0.072 | 0.191 |
| 9 | 3.1 | 4.5+14=18.5 | 0.043 | 0.058 | 0.014 | 0.044 |
| 10 | 3.1 | 5+14=19 | 0.044 | 0.059 | 0.016 | 0.044 |
| 11 | 3.0 | 10+18=28 | 0.053 | 0.074 | 0.026 | 0.048 |
| 12 | 3.0 | 16+40=56 | 0.106 | 0.150 | 0.043 | 0.107 |
| 13 | 3.6 | 5+5=10 | 0.057 | 0.080 | 0.040 | 0.040 |
| 14 | 3.3 | 9+6.6=15.6 | 0.052 | 0.103 | 0.059 | 0.044 |
| 15 | 3.2 | 14.5+12.9=27.4 | 0.076 | 0.156 | 0.083 | 0.073 |
| 16 | 2.7 | 3+54=57 | 0.057 | 0.095 | 0.005 | 0.090 |
| 17 | 3.0 | 20+40=60 | 0.114 | 0.150 | 0.050 | 0.100 |

Note that since the ions were not separated by mass on the Explorer 45 satellite, the values of
the full energy density of ions given in the first two rows of Table 2 are underestimated
significantly and mainly correspond to protons: at equal fluxes and energies, the energy density of
O$^+$ ions is 4 times higher compared with protons.
For some of the storms considered here, simultaneous measurements of the magnetic field at the
RC maximum are given. Using the measurements of the magnetic field on the Explorer 45 satellite
during the storm on December 17, 1971 (Anderson and Gurnett, 1973), for the first row of Table 2,
we get $\beta_m = 0.188$ (instead of the value 0.200, calculated by the general formula); during the storm
on February 24, 1972 (Cahill and Lee, 1975), for the second row of Table 2, we get $\beta_m = 0.136$
(instead of the value 0.139). Using the measurements of the magnetic field on the AMPTE/CCE
satellite during the storm on September 5, 1984 (Potemra et al., 1985; Krimigis et al., 1985), for
the fourth row of Table 2, we get $\beta_m = 0.260$ (instead of 0.243). The values of the parameter $\beta_m$ in
the first, second, and fourth rows of Table 2 have been corrected taking into account this remark.
The values of the parameter $\beta_m$ from Table 2 at the maximum of the RC are shown in Fig. 2 in
the $\{\beta_m, |D_{st}|\}$, $\{\beta_m, \text{MLT}\}$, and $\{\beta_m, L_m\}$ spaces. Here we present only the results that were
obtained at the end of the main phase of storms and refer to the night magnetosphere.
For the points 13, 16 and 17, which belong to 03–05 MLT sector, the values of the parameter $\beta_m$
are overestimated significantly, because on the main phase of storms, the magnetic field depression
in this sector is insignificant. For these points in Fig. 2*a* parameter $\beta_{md}$ is used.
For the point 5, the value of parameter $\beta_m$ exceeds significantly the other values of $\beta_m$ in Fig. 2.
This point belongs to 18 MLT and was obtained at the end of the main phase of a very complex,
multistage superstorm.

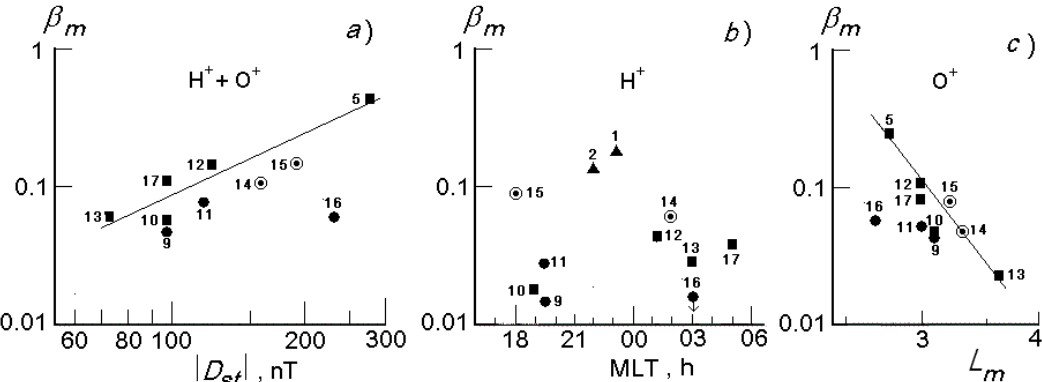

**Figure 2.** Ratio of the energy densities for ions and magnetic field at the maximum of the RC ($\beta_m$) depending on $|D_{st}|$ (a), MLT (b), and on $L_m$ (c). The thin lines are mean-square approximations of these data.

Figure 2a represents the distribution of parameter $\beta_m$, corresponding to the energy density of $H^+ + O^+$ ions, in $\{\beta_m, |D_{st}|\}$ space. This figure shows that for ions with $E \sim 1–300$ keV (the points 5, 10, 12, 13, 14, 15, and 17), parameter $\beta_m$ increases with the increasing $|D_{st}|$; this dependence is approximated by the following expression (thin line in Fig. 2a):

$$\beta_m = 8.6 \cdot 10^{-5} |D_{st}|^{1.494}$$

with correlation coefficient $R = 0.866$. Here $D_{st}$ is in nT.

It can be seen from Table 2, that compare with protons, for $O^+$ ions with $E \sim 1–300$ keV, the scatter in the values of the parameter $\beta_m$ is smaller; however, for $O^+$ ions $\beta_m$ practically does not correlate with $|D_{st}|$.

Figure 2b represents the distribution of parameter $\beta_m$ for $H^+$ ions in the $\{\beta_m, \text{MLT}\}$ space. This figure shows that for ions with $E \sim 1–300$ keV (the points 1, 2, 10, 12, 13, 14, and 15) parameter $\beta_m$ increases in the evening sector and decreases in the morning sector with an increase in MLT. The points 14 and 15, which refer to the end of a very irregular and long ($\sim 17$ h) main phase of a strong storm on March 17, 2015, reflect here the symmetric component of the ion RC (50–200 keV).

It can be seen from Table 2, that for $O^+$ ions, as well as for $H^+ + O^+$ ions, much more complex, irregular distributions are obtained in the $\{\beta_m, \text{MLT}\}$ space.

Figure 2c represents the distribution of parameter $\beta_m$ for $O^+$ ions in the $\{\beta_m, L_m\}$ space. From this figure it can be seen that parameter $\beta_m$ increases with a decrease in $L_m$. For ions with $E \sim 1–300$ keV (the points 5, 10, 12, 13, 14, 15, and 17), we obtain the following dependence (thin line in Fig. 2c):

$$\beta_m = 2.82 \cdot 10^3 \cdot L_m^{-9.232.}$$

with correlation coefficient $R = -0.866$.

It can be seen from Table 2, that for $H^+$ ions, as well as for $H^+ + O^+$ ions, a very chaotic distributions are obtained in the $\{\beta_m, L_m\}$ space.

## 3.3 Parameters of the ionic ring current on $L > L_m$

During the main phase of storms, the inner edge of the ionic RC is very steep: as $L$ shells decreases from $L_m$ to $L_m - \Delta L$, the ionic RC energy density decreases by an order of magnitude at $\Delta L/L_m \sim$ 0.2–0.3 (see, e.g., Krimigis et al ., 1985; McEntire et al., 1985; Hamilton et al., 1988; Greenspan and Hamilton, 2000, 2002; Gkioulidou et al., 2014; Kistler et al., 2016; Menz et al., 2017; Keika et





al., 2018); in most problems associated with the simulation of the RC, the shape of its inner edge
does not play a significant role. At the same time, the outer part of the RC (for $L > L_m$) has a much
smaller gradient.
According to the results of the experiments indicated in Table. 1, the radial dependences of the
RC ions energy density $w(L)$ at $L > L_m$ are well approximated by an exponential function:
$$w(L) = w_0 \exp(-L/L_0).$$
The parameter $w_0$ characterizes the intensity of the RC, and the parameter $L_0$ characterizes its
steepness on $L > L_m$.
The parameters $w_0$ and $L_0$ of the RC were calculated by the least squares method for each
experimental profile $w(L)$, separately for $H^+$ and $O^+$ ions, and also for their total ($H^+ + O^+$) energy
density. For ions of low and high energies (in the ranges of different widths), the results of these
calculations were considered separately.
The correlation coefficients $R$ of such approximations with experimental data are very high;
thus, for the total energy density of ions ($H^+ + O^+$), it ranges from $-0.812$ to $-0.999$, and for most
of the measurements considered here, $R < -0.96$. When these dependences are approximated by
other simple functions (for example, a power function), much weaker correlation coefficients are
obtained.
The results of these calculations are given in Table 3. The first column of this table corresponds
to the first column of Tables 1 and 2. The second and third columns present the intervals $L$ and
MLT, for which these parameters were calculated. The fourth column presents the values of $|D_{st}|$
corresponding to these measurements. The remaining columns of this table presents the values of
parameters $w_0$ and $L_0$ for $H^+ + O^+$ ions and separately for $H^+$ and $O^+$ ions. The rows in Table 3
correspond to the rows in Tables 1 and 2.
From the data given in (Yue et al., 2018) and corresponding to row 13 in Tables 1 and 2, it is
possible to reliably determine the RC parameters at its maximum, but parameters of the RC in its
outer part are determined with large errors; therefore, for this storm, parameters of the outer part of
the RC are not presented in Table 3.

**Table 3**

| | $L$ | MLT, h | $|D_{st}|$, nT | $w_0$, nPa ($H^+ + O^+$) | $L_0$ ($H^+ + O^+$) | $w_0$, nPa ($H^+$) | $L_0$ ($H^+$) | $w_0$, nPa ($O^+$) | $L_0$ ($O^+$) |
|---|---|---|---|---|---|---|---|---|---|
| 1 | 3.5–5.0 | 21.30–23.00 | 158–171 | – | – | 230 | 2.14 | – | – |
| 2 | 3.5–5.0 | 19.30–22.00 | 53–83 | | – | 66 | 2.94 | – | – |
| 3 | 4.5–7.5 | 10.40–12.20 | 49–51 | 72 | 2.28 | 21.5 | 2.83 | 65 | 1.74 |
| 4 | 3.5–6.5 | 15.40–17.40 | 69–73 | 496 | 1.46 | 337 | 1.58 | 326 | 1.27 |
| 5 | 3.0–5.0 | 14.00–17.00 | 259–266 | 1079 | 1.13 | 995 | 1.19 | 1880 | 0.95 |
| 6 | 3.5–5.0 | 3.00–4.30 | 38–69 | 652 | 1.39 | – | – | – | – |
| 7 | 3.5–5.5 | 19.30–22.30 | 66–86 | 673 | 1.13 | 19 | 3.47 | 1032 | 0.96 |
| 8 | 3.5–5.5 | 20.00–22.00 | 66–85 | 443 | 1.34 | 35 | 2.97 | 329 | 1.35 |
| 9 | 3.5–4.5 | 20.00–21.30 | 98–115 | 311 | 1.09 | 21 | 1.70 | 524 | 0.86 |
| 10 | 3.5–5.5 | 19.50–22.30 | 98–123 | 102 | 1.89 | 70 | 1.79 | 43 | 1.86 |
| 11 | 3.0–5.0 | 19.30–22.30 | 115–132 | 203 | 1.50 | 55 | 1.67 | 98 | 1.75 |
| 12 | 3.5–5.5 | 23.00–01.00 | 109–115 | 368 | 1.82 | 157 | 1.60 | 429 | 1.50 |
| 14 | 3.5–5.5 | 00.00–02.00 | 166–180 | 95 | 1.89 | 104 | 1.44 | 134 | 1.20 |
| 15 | 3.5–5.0 | 18.00–20.20 | 190–216 | 214 | 1.62 | 52 | 2.88 | 1614 | 0.74 |
| 16 | 3.0–5.0 | 00.00–02.00 | 198–233 | 87 | 1.66 | – | – | – | – |
| 17 | 3.0–6.0 | 05.00–07.00 | 88–98 | 348 | 1.61 | 100 | 1.86 | 508 | 1.18 |



Figure 3 shows the approximation dependences of $\ln w$ on $L$ for $L > L_m$, where $w(L)$ is the
energy density of $H^+ + O^+$ ions (in nPa). In this figure, ions with $E \sim 1\text{--}300$ keV are represented by
dark segments, and ions with $E \sim 1\text{--}60$ keV are represented by red segments. The numbers at the
beginning and at the end of these segments correspond to the lines in Table 1–3.

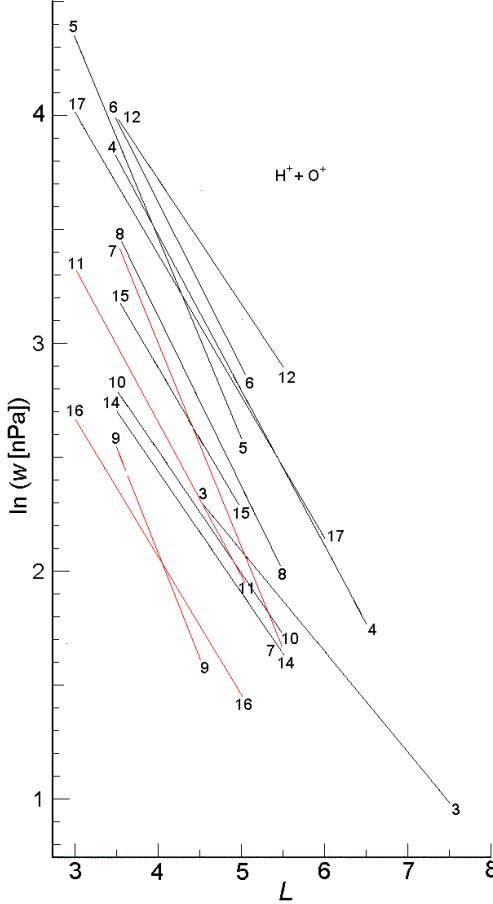


**Figure 3.** Radial profiles of the energy density ($w$) of $H^+ + O^+$ ions with $E \sim 1\text{--}300$ keV (dark lines), and with $E \sim 1\text{--}60$
keV (red lines) for the outer part of the RC on the main phase of various storms.

Figures 4 and 5 presents the distributions of $w_0$ and $L_0$ parameters (from Table 3) depending on
$|D_{st}|$, MLT and $L_m$. These figures show the results that refer to the end of the main phase of storms
(except for the point 7 in Fig. 4) and were obtained in the evening and near midnight sectors of
MLT (except for the point 5). These results refer to the energy density of $H^+ + O^+$ ions (except for
the points 1 and 2, which mainly refer to protons). These distributions are depends very strongly on
the energy range of the ions, which leads to a large scatter of the points in these figures.
When a satellite crosses the RC region, the values of $L$, $|D_{st}|$, and MLT are changes. Positions of
the points on Figs. 4–6 corresponds to the average values of $|D_{st}|$ and MLT when the satellites
crosses the outer part of the RC (horizontal segments indicate changes in $|D_{st}|$ and MLT during
these periods).
Figure 4$a$ presents the distribution of $w_0$ in the $\{w_0, |D_{st}|\}$ space. From this figure it can be seen
that for $H^+ + O^+$ ions with $E \sim 1\text{--}300$ keV the value of $w_0$ increases, while for the ions with $E \sim 1\text{--}$
60 keV it is decreases with increase in $|D_{st}|$.

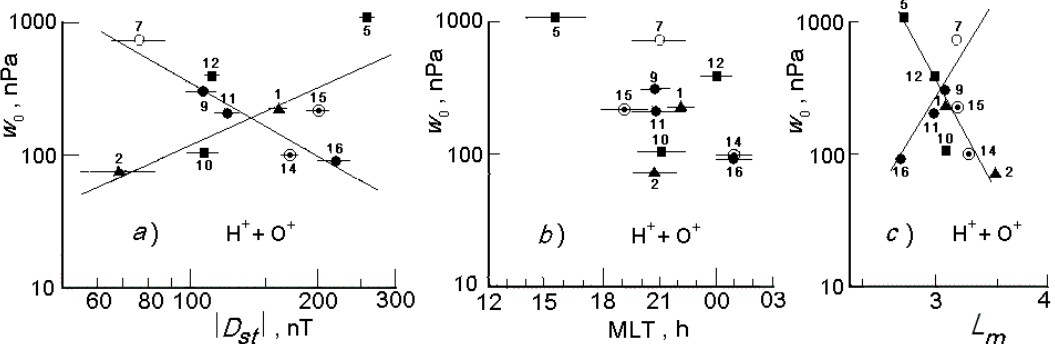


**Figure 4**. Distributions of $w_0$ depending on $|D_{st}|$ (*a*), MLT (*b*), and $L_m$ (*c*). These results refer to the energy density of $H^+ + O^+$ ions (except for the points 1 and 2, which mainly refer to protons), and belong to the end of the main phase of storms (except for the point 7). Thin lines show mean-square approximations of these distributions.

For the set of points 1, 2, 5, 10, 12, 14, and 15 in Fig. 4*a*, which refer to ions with $E \sim 1$–300 keV, we obtain the following least squares approximation (ascending line in Fig. 4*a*):

$$w_0 = 0.143 \cdot |D_{st}|^{1.457}$$

with correlation coefficient $R = 0.696$. Here $w_0$ is in nPa, and $D_{st}$ is in nT.

For the set of points 7, 9, 11, and 16 in Fig. 4*a*, which refer to ions with $E \sim 1$–60 keV, we obtain the following least squares approximation (descending line in Fig. 4*a*):

$$w_0 = 5.0 \cdot 10^5 \, |D_{st}|^{-1.615}$$

with correlation coefficient $R = -0.999$. Here $w_0$ is in nPa, and $D_{st}$ is in nT.

Note that for the point 7, according to the Van Allen Probes B satellite, the radial profiles of energy density for ions with $E = 10$–60 keV and 10–570 keV were almost identical (see Fig. 3 in Menz et al., 2017). This means that during this period, in the 19.30–22.30 MLT sector, the ions with $E = 10$–60 keV was provided almost full contribution to the energy density of the RC.

Thus, parameter $w_0$ for $H^+ + O^+$ ions with $E \sim 1$–300 keV correlates, and for ions with $E \sim 1$–60 keV it anticorrelates with $|D_{st}|$, i.e., the stronger storm, the smaller the fraction of low-energy ions and the larger the fraction of high-energy ions in the total energy density of the RC (the average kinetic energy of the ions increases). When $H^+$ and $O^+$ ions are considered separately, this effect manifests itself for $H^+$ ions, and does not appear for $O^+$ ions.

Figure 4*b* presents the distribution of parameter $w_0$ in the $\{w_0, \text{MLT}\}$ space. This figure shows that, on the main phase of storms, the values of $w_0$ in the evening and near midnight sectors of MLT have a very large scatter.

Figure 4*c* presents the distribution of parameter $w_0$ in the $\{w_0, L_m\}$ space. From this figure it can be seen that for $H^+ + O^+$ ions with $E \sim 1$–300 keV parameter $w_0$ increases, while for ions with $E \sim 1$–60 keV it decreases with decreasing $L_m$.

For the set of points 1, 2, 5, 10, 12, 14, and 15 in Fig. 4*c*, which refer to ions with $E \sim 1$–300 keV, the following approximation was obtained by the least squares method (descending line in Fig. 4*c*):

$$w_0 = 2.39 \cdot 10^8 \, L_m^{-12.246.}$$

with correlation coefficient $R = -0.910$. Here $w_0$ is in nPa.

For the set of points 7, 9, 11, and 16 in Fig. 4*c*, which refer to ions with $E \sim 1$–60 keV, the following approximation was obtained by the least squares method (ascending line in Fig 4*c*):




$$w_0 = 1.13 \cdot 10^{-3} \, L_m^{11.213}$$
with correlation coefficient $R = 0.989$. Here $w_0$ is in nPa.
Thus, parameter $w_0$ for $H^+ + O^+$ ions with $E \sim 1$–60 keV correlates, and for ions with $E \sim 1$–300
keV it anticorrelates with $L_m$, i.e., the closer the RC approaches to the Earth, the smaller the
fraction of low-energy ions and the larger the fraction of high-energy ions in the total energy
density of the RC (the average kinetic energy of the ions increases). When $H^+$ and $O^+$ ions are
considered separately, this effect manifests itself for $H^+$ ions and does not appear for $O^+$ ions.
Figure 5 presents the distributions of parameter $L_0$ in the $\{L_0, |D_{st}|\}$, $\{L_0, \text{MLT}\}$, and $\{L_0, L_m\}$
spaces. In contrast to parameter $w_0$, in the experimental results considered here for ions with $E \sim$
1–60 keV, the distributions of parameter $L_0$ are much less ordered and there are no clear
regularities in them. Therefore, Fig. 5 show the results only for the energy density of $H^+ + O^+$ ions
with $E \sim 1$–300 keV. The points 1 and 2, which mainly refer to protons, are given here for
comparison.

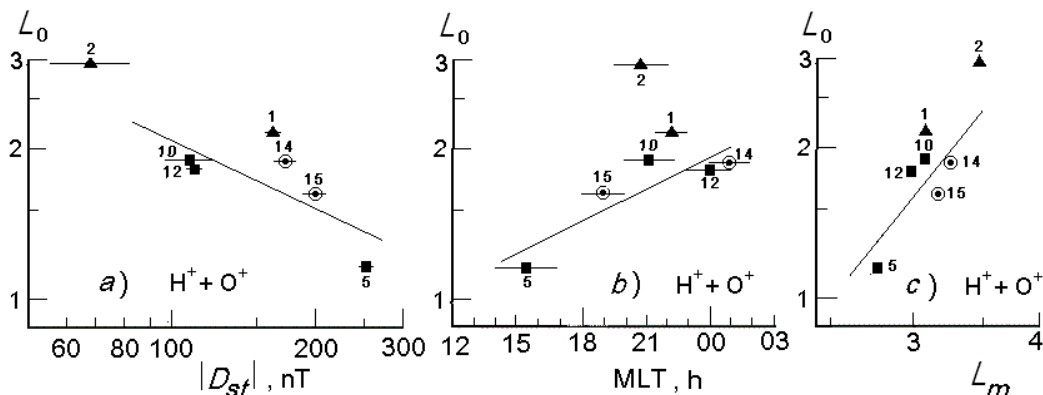


**Figure 5**. Distributions of parameter $L_0$ depending on $|D_{st}|$ (*a*), MLT (*b*), and $L_m$ (*c*). These results refer to the energy
density of $H^+ + O^+$ ions with $E \sim 1$–300 keV (the exception is only for the points 1 and 2, which mainly refer to protons)
and were obtained at the end of the main phase of storm. Thin lines show mean-square approximations of these
distributions.
Figure 5*a* presents the distributions of parameter $L_0$ in the $\{L_0, |D_{st}|\}$ space. From this figure it
can be seen that for $H^+ + O^+$ ions with $E \sim 1$–300 keV in the evening and near midnight sectors of
MLT, the average value of parameter $L_0$ decreases with an increase in $|D_{st}|$. For the set of points 5,
10, 12, 14, and 15, we obtain the following least squares approximation (thin line in Fig. 5*a*):
$$L_0 = 17.9 \, |D_{st}|^{-0.469}$$
with correlation coefficient $R = -0.814$. Here $w_0$ is in nPa.
Figure 5*b* presents the distribution of parameter $L_0$ in the $\{L_0, \text{MLT}\}$ space. This figure
demonstrates the strong azimuth asymmetry of the RC on the main phase of storms; for $H^+ + O^+$
ions with $E \sim 1$–300 keV parameter $L_0$ is maximum in the sector $\sim 21$–24 MLT. For the set of
points 5, 10, 12, 14, and 15, we obtain the following least squares approximation (thin line in Fig.
5*b*):
$$L_0 = 0.626 \cdot \exp(\text{MLT}/20.14)$$
with correlation coefficient $R = 0.882$. Here MLT is expressed in hours.
Figure 5*c* presents the distribution of parameter $L_0$ in the $\{L_0, L_m\}$ space. From this figure it can
be seen that for $H^+ + O^+$ ions with $E \sim 1$–300 keV, the closer the RC approaches to the Earth, the





steeper its outer part. For the set of points 5, 10, 12, 14, and 15, we obtain the following least
squares approximation (thin line in Fig. 5*c*):
$$L_0 = 8.79 \cdot 10^{-2} L_m^{2.606}$$

with correlation coefficient $R = 0.774$.
If the $H^+$ and $O^+$ ions are considered separately, it can be concluded that parameters $w_0$ and $L_0$
correlate with $D_{st}$, MLT, and $L_m$ much worse, especially for $O^+$ ions. For example, Fig. 6 presents,
separately for $H^+$ and $O^+$ ions, the distributions of parameters $w_0$ and $L_0$ by MLT at the end of the
main phase of storms. In these distributions, the correlation of parameters $w_0$ and $L_0$ with MLT is
very weak; for $H^+$ ions this correlation are better than for $O^+$ ions.
During the storm in February 1986 (the point 5 in our figures), the AMPTE/CCE satellite orbit
crossed the RC region mainly in the daytime and only at $L < 3$ it pass in the evening sector
(Hamilton et al., 1988). In some distributions shown in Figs. 4–6, the point 5 deviates from the
general trends; the most significant deviations of this point were obtained for Fig. 6 (the point 5 is
excluded from this figure).

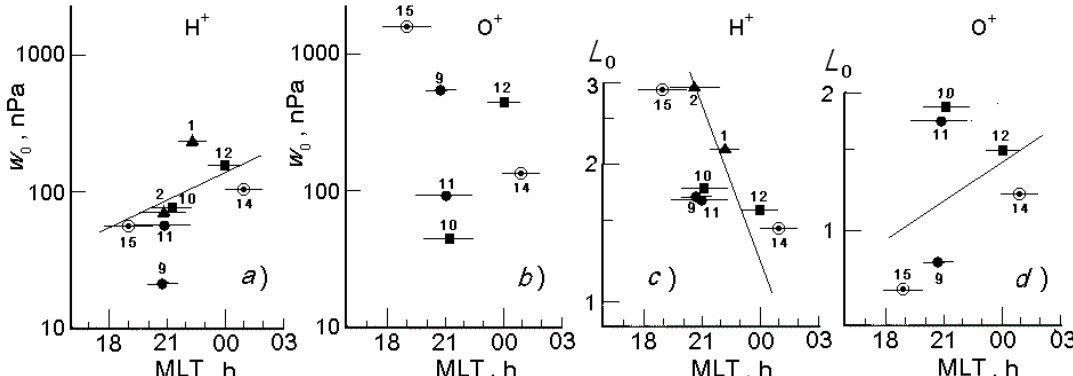

**Figure 6**. Distributions of parameters $w_0$ and $L_0$ by MLT, plotted separately for $H^+$ and $O^+$ ions (see text).
For the set of points 1, 2, 10, 12, 14, and 15 in Fig. 6*a*, we obtain the following least squares
approximation for $H^+$ ions with $E \sim 1$–300 keV (thin line in Fig. 6*a*):
$$w_0 = 2.625 \cdot \exp(MLT/6.086)$$

with correlation coefficient $R = 0.622$. Here MLT is expressed in hours.
On Fig. 6*b*, the experimental points are not correlated with MLT.
For the set of points 10, 12, 14, and 15 in Fig. 6*c*, we obtain the following least squares
approximation for $H^+$ ions with $E \sim 1$–300 keV (thin line in Fig. 6*c*):
$$L_0 = 710 \cdot \exp(-MLT/3.77)$$

with correlation coefficient $R = -0.882$. Here MLT is expressed in hours.
For the set of points 10, 12, 14, and 15 in Fig. 6*d*, we obtain the following least squares
approximation for $O^+$ ions with $E \sim 1$–300 keV (thin line in Fig. 6*d*):
$$L_0 = 0.291 \cdot \exp(MLT/15.28)$$

with correlation coefficient $R = 0.440$. Here MLT is expressed in hours.
From the experimental results considered here, one can see also that the average values of
parameters $L_0$ and $w_0$ of the RC increases with an increase in the rate of change $|D_{st}|$ on the main
phase of the storms. However, these correlations are very weak.



Thus, the distributions presented in Figs. 1–6 give a fairly complete general description of the
structure and dynamics of the ion RC on the main phase of geomagnetic storms. These
distributions make it possible to identify some clear regularities for the main parameters of the ion
RC. At the same time, it is important to note the large scatter of the experimental points in these
figures, which is caused by the complex and partly non-universal nature of the dynamics of the RC
and the magnetosphere as a whole on the main phase of storms.
**4 Discussion**
**4.1 Region near the maximum of the energy density of the ring current**
The performed analysis of the experimental data shows that the position of the maximum the
energy density of RC ions ($L_m$) clearly anticorrelates with the value of $|D_{st}|$, despite that these
storms had not only different intensities, but also different character of variations of the $D_{st}$. This
indicates a connection between quantities $L_m$ and $D_{st}$, which should be provided by the physical
mechanism, which is universal for the main phase of storms. On the main phase of storms with $D_{st}$
$< -50$ nT, such mechanism is the convection of the RC ions, drifting in the Earth's magnetosphere
under the action of large-scale magnetic and electric fields with conservation of the first ($\mu$) and
second ($K$) adiabatic invariants (see, e.g., Ebihara and Ejiri, 2003).
When analyzing the development of the RC on the main phase of storms, one should also take
into account substorms and the rapid variations in the electric and magnetic fields in the outer
regions of the geomagnetic trap, which lead to more effective acceleration and transport of the ions
(see, e.g., Fu et al., 2001, 2002; Ganushkina et al., 2005; Gkioulidou et al., 2014, 2015; Thaller et
al., 2015; Nosé et al., 2016; Keika et al., 2016; Mitchell et al., 2018).
The experimental data show that during the main phase of storms, the electric and magnetic
fields vary greatly (in the range from minutes to tens of minutes), especially in the outer part of the
geomagnetic trap (see, e.g., Yang et al., 2016). Therefore, the real drift trajectories of separate ions
are irregular, and the large-scale convection of RC ions must be considered as a time-averaged
pattern (see, e.g., Chen et al., 1994).
At the same time, deep penetration of the RC ions into a geomagnetic trap is possible only
during the periods of strong hot plasma convection on the main phase of storms under the action of
quasi-stationary fields (Daglis et al., 1999; Kovtyukh, 2001).
Let us consider a simple model of such convection for $H^+$ and $O^+$ ions, which make the primary
contribution to the energy density of the RC particles on the main phase of storms. If we do not
take into account the loss of ions, then at the same energies and pitch angles (at the same $\mu$ and $K$),
the drift trajectories of these ions are identical.
In the region of the near plasma sheet of the magnetotail, ions drift towards the Earth in crossed
magnetic and electric fields. Reaching the region of the quasi-dipole magnetic field, these ions drift
to the Earth under the action of the electric field of convection and gradually deviate to the east,
into the morning sector, under the action of electric fields of convection and corotation. In the
region of the quasi-dipole and dipole magnetic fields, the ion magnetic drift velocities are directed
to the west and, under conserving $\mu$ and $K$, they increases as the ions approach to the Earth, while
the ion electric drift velocities decreases.
As a result, the ratio of velocity of the magnetic to electric drifts of ions with a certain values of
$\mu$ and $K$ increases, and at some dot in the morning sector the magnetic drift overpowers the electric
drift; at this dot the ions begin to drift to the west, continuing to increase their energy. In the
evening sector the ions reach their maximum energies, and then drift towards the noon and late
morning sectors, losing their energy, and, under the action of the electric fields of convection and
corotation, turn to the east.
Carrying out consideration in the equatorial plane, for the ions with an equatorial pitch angle $\alpha_0$
$\sim 90^o$ ($K \approx 0$), we will assume, for simplicity, that in the $L_m \sim 2.5$–3.5 region the geomagnetic field
is close to the dipole configuration ($B \propto L^{-3}$). Unlike the electric field, the approximation of the

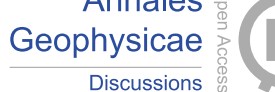

geomagnetic field in the trap to the real configuration does little to change the pattern of ion drift (and the results of mathematical modeling of RC) during storms (see, e.g., Menz et al., 2019a).

It can be assumed that parameter $L_m$ of the RC on the main phase of storms corresponds to the dot of reversal of the drift trajectories of ions with some average value of $\bar{\mu}$. At this dot, the velocities of the electric and magnetic drifts of ions (see, e.g., Roederer, 1970) directed towards each other and mutually cancel:

$$32 \cdot k \, |\mathbf{E}| \, L_m^3 + 464 \cdot L_m \approx 472 \cdot 10^3 \cdot \bar{\mu} \, L_m^{-1} \,, \tag{1}$$

where $|\mathbf{E}|$ (mV/m) is the electric field strength of the convection near the maximum of the RC, the coefficient $k$ determines the azimuthal projection of the vector $\mathbf{E}$ ($k \sim 0.5$–$1.0$), and $\bar{\mu}$ (keV/nT) is the average value of the first adiabatic invariant of the RC ions. The left side of Eq. (1) contains the eastward drift velocity of ions under the action of electric fields of convection and corotation (in m/s), and the right side is the westward magnetic drift velocity of ions (in m/s).

According to such view, on the main phase of storms for any RC ions with $Q_i = +1$ (in particular, for H$^+$ and O$^+$ ions), the maxima of energy density of these ions should be close in $L$, which is confirmed by many measurements (see, e.g., Krimigis et al., 1985; McEntire et al., 1985; Stüdemann et al., 1986; Hamilton et al., 1988; Greenspan and Hamilton, 2002; Kistler et al., 2016; Menz et al., 2017; Keika et al., 2018: Yue et al., 2018).

On the main phase of storms, the value of $D_{st}(T)$ is proportional to the integral by the electric field strength $|\mathbf{E}|$ over time from the beginning of the storm to the moment $T$ (Burke et al., 2007); after averaging over 17 storms with max$|D_{st}|$ from $-100$ to $-470$ nT, the following relation was obtained (with a correlation coefficient of 0.93):

$$D_{st} = 7.3 - 24.1 \int_0^T |\mathbf{E}| \, dt, \tag{2}$$

where $D_{st}$ is in nT, $\mathbf{E}$ is in mV/m, and $T$ is in hours.

For strong storms, we can neglect the constant 7.3 (nT) on the right side of Eq. (2), and the integral can be replaced on the average value $\langle|\mathbf{E}|\rangle$ multiplied by $T$ (see Fig. 3 in Burke et al., 2007):

$$D_{st} \approx -24.1 \langle|\mathbf{E}|\rangle T \,. \tag{2a}$$

The values $|\mathbf{E}|$ were determined in Burke et al. (2007) by dividing the potential difference across the polar cap by the transverse size of the magnetosphere, and by this method it was obtained: $\langle|\mathbf{E}|\rangle \sim 1$ mV/m. However, from the data of the satellites CRRES (Wygant et al., 1998; Burke et al., 1998; Korth et al., 2000; Garner et al., 2004), Akebono (Nishimura et al., 2006, 2007), and Van Allen Probes (Thaller et al., 2015), on the main phase of strong storms, near the RC maximum (at $L \sim 3$–$4$) in the evening and near-midnight MLT the value of $|\mathbf{E}|$ can achieve $\sim 2$–$10$ mV/m.

Thus, during the storm on March 24, 1991 (max$|D_{st}| = 298$ nT), the convection electric field was penetrated up to $L \sim 2$ and achieved 8 mV/m, while at $L > 4$ it did not exceed 1–2 mV/m; at the end of the main phase of this storm, the maximum depression of the magnetic field in the trap associated with RC was localized at $L = 2.4$, in the same place as the electric field maximum (Wygant et al., 1998). On the main phase of this storm, such strong electric fields were persisted for several hours and could inject ions from $L = 8$ to $L = 2.4$; in this case, the ions can be adiabatically accelerated from 1–5 keV to 300 keV.

These experimental results are explained by two interrelated physical effects near the RC maximum, which are inherent for convection on the main phase of strong storms: Subauroral Polarization Streams, SAPS (see, e.g., Garner et al., 2004) and Subauroral Ion Drifts, SAID (see, e.g., Wang et al., 2021).



Therefore, for the region of the RC maximum, the coefficient on the right side of Eq. (2a) need
to be reduced; when it is reduced by 6 times, Eq. (1) takes the following form:

$$8 \frac{k}{T} |D_{st}| L_m^4 + 464 L_m^2 - 472 \cdot 10^3 \, \overline{\mu} \approx 0 . \tag{3}$$

This equation has a unique positive solution:

$$L_m \approx \left( \sqrt{59 \cdot 10^3 \, \overline{\mu} \, T / k |D_{st}| + \left( 29 \, T / k |D_{st}| \right)^2} - 29 \, T / k |D_{st}| \right)^{0.5} . \tag{4}$$

The value of $\overline{\mu}$ approximately corresponds to the maximum of the differential energy density of
the RC ions. On the data from Van Allen Probes for H$^+$ and O$^+$ ions, this maximum corresponds to
~ 0.05–0.07 keV/nT (see, e.g., Mentz et al., 2017; Keika et al., 2018).
For almost all the storms considered here, the time from the beginning of the storm to the
moment of the RC measurements on the main phase of the storm corresponds the range $T \sim 2$–12
h; only the storm in February 1986 falls out from this range (row 5 in Table 1), for which $T \sim 23$ h.
The first term under the radical in Eq. (4) exceeds significantly the second term at $|D_{st}| \sim 100$–
300 nT, $T \sim 2$–12 h (usually, as $|D_{st}|$ increases, parameter $T$ increases also), and $\overline{\mu} \sim 0.05$–0.07
keV/nT. Therefore, Eq. (4) can be simplified:

$$L_m \approx \left( \sqrt{59 \cdot 10^3 \, \overline{\mu} \, T / k |D_{st}|} - 29 \, T / k |D_{st}| \right)^{0.5} . \tag{5}$$

This result corresponds to the fact that for the values of $L_m$, $|D_{st}|$, and $\overline{\mu}$ considered here, the
magnetic drift velocity of singly charged ions exceeds significantly the corotation rate. However,
the corotation plays an important role in the overall balance of the ion drift velocities.
It is illustrate Fig. 7, which was constructed for $\overline{\mu} = 0.07$ keV/nT, $T = 10$ h, and $k = 0.75$. In this
figure, the thin black line represents Eq. (4), and the red line represents Eq. (5); it can be seen that
these lines are very close to each other, and in the range of $|D_{st}| \sim 100$–300 nT they are almost
identical. The dotted line in this figure also shows the curve obtained for the same values of $\overline{\mu}$, $T$,
and $k$, if the corotation of the RC ions is neglected completely.

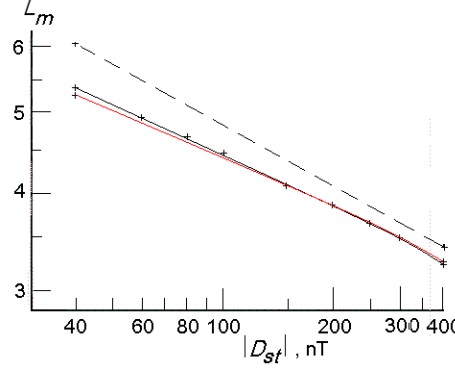

**Figure 7**. Comparison of Eqs. (4) and (5) obtained for a simple model of the RC ions convection (see text).
Figure 1a shows, as red curves, the dependences $L_m(|D_{st}|)$ calculated by Eq. (5) for $k = 1$: the
lower curve corresponds to $\overline{\mu} = 0.05$ keV/nT and $T = 2$ h, and the upper curve corresponds to $\overline{\mu} =$
0.07 keV/nT and $T = 12$ h. In our model, these curves fits to the minimum and maximum values of
parameter $L_m$ (in the ranges of values $\overline{\mu}$ and $T$ considered here).
Taking into account the weakening of the magnetic field at $L \sim L_m$ during storms (see Sect. 3.2),





as $L_m$ decreases, the average energy of ions corresponding to the lower red curve ($\overline{\mu} = 0.05$
keV/nT) increases from ~ 44 keV at $L_m = 3.1$ to ~ 62 keV at $L_m = 2.7$, and for the upper red curve
($\overline{\mu} = 0.07$ keV/nT) this energy increases from ~ 39 keV at $L_m = 3.6$ to ~ 70 keV at $L_m = 2.8$.
The red dotted line in Fig. 1$a$, for the same values of parameters $\overline{\mu}$, $T$, and $k$, the curves are also
shown, which are obtained if the RC ions corotation is fully neglected (if the second term in Eq.
(3) is equated to zero). In this case, power-law dependences are obtained, which give higher values
of parameter $L_m$:

$$L_m \approx \left( 59 \cdot 10^3 \, \overline{\mu} \, T / k \right)^{0.25} |D_{st}|^{-0.25}. \qquad (6)$$

In the outer region of the RC (at $L > L_m$), the influence of corotation on the pattern of
convection of the RC ions increases with an increase in $L$ shell. During the main phase of storms,
corotation make for the closure of the drift trajectories of ions into asymmetric loops of the partial
ring current.
The experimental dependence $L_m(|D_{st}|)$ is about in the middle of the range limited in Fig. 1$a$
with thin red lines. This range is mainly determined by the width of the interval for the parameter
$T$. As $|D_{st}|$ increases, the experimental values of $L_m$ are removed from the lower boundary of this
range ($T = 2$ h) and approach to its upper boundary ($T = 12$ h), in accordance with the fact that for
most of the storms considered here, the average value of parameter $T$ increases with increase $|D_{st}|$.
The simple convection model considered here also explain the fact that in the evening and near-
midnight sectors parameter $L_m$ is practically independent from MLT (see Fig. 1$b$): on $L \sim L_m$, the
magnetic drift of ions to the west dominates, and in the dipole field the trajectories of this drift are
close to circles concentric with the Earth.
It is interesting to compare Fig. 1 with Fig. 2.
Parameter $L_m$ decreases (Fig. 1$a$), and parameter $\beta_m$ increases (Fig. 2$a$) with an increase in
intensity of magnetic storms. These results are evidenced to the acceleration of the RC ions.
The experimental results shown in Fig. 1$b$ do not give a systematic dependence of the RC
parameter $L_m$ on MLT. However, Fig. 2$b$ shows that for H$^+$ ions parameter $\beta_m$ strongly depend on
MLT and reaches its maximum values in the pre-midnight sector. At the same time, for O$^+$ ions
there is no the systematic dependence of $\beta_m$ on MLT.
Figure 2$c$ shows that when the RC maximum shifts towards the Earth, parameter $\beta_m$ for O$^+$ ions
increases (there is no such correlation for H$^+$ ions).
Thus, these comparison of Figs. 1 and 2 show that the RC parameters depend not only on the
electric and magnetic fields and their variations, but also on factors related to the nature and origin
of the ions themselves. These factors include the shape of the energy spectra and spatial
distributions of ions in their source, as well as the loss rates of ions during their convection (all
these factors differ significantly for H$^+$ and O$^+$ ions).
Let's take a closer look at Fig. 2$c$.
On the main phase of storms, ions drift from the plasma sheet (PS) of the magnetotail to the
Earth with the conservation of the first adiabatic invariant $\mu$ and, therefore, the kinetic energy $E$ of
near-equatorial particles increases as $\mu B$ (this is confirmed by numerous experimental data and
their comparison with the results of mathematical modeling of the RC). In this case, the interval
$\Delta\mu$, corresponding to the interval $\Delta E$ fixed by the instrument on the satellite, shifts to the smaller $\mu$
values. In addition, in the absence of significant energy loss of particles, the ion fluxes ($J$) of the
corresponding energies ($\mu = const$) along their drift trajectories change in accordance with the
Liouville theorem ($J/B = const$).
Therefore, the energy density of ions $w$ in a fixed interval $\Delta E$ must increases with a decrease in
$L$; this dependence is the stronger, the steeper the boundary energy spectrum of the considered ions
in their source. As a result, for the dipole magnetic field (at $L_m < 3.5$), and for the power-law



approximation of the boundary differential energy spectrum of the ion fluxes ($J \propto E^{-\gamma}$), we obtain
the following dependence: $w_m \propto B^{\gamma+1}$, where $B = B(L_m)$ at the equatorial plane, or $w_m \propto L_m^{-3(\gamma+1)}$.
In the range $\mu \sim 0.01$–$0.2$ keV/nT (it is correspond to ions with $E \sim 10$–$300$ keV at $L \sim 3.0$–$3.5$)
the average energy spectrum of $O^+$ ions in the PS region adjacent to the RC has the exponent $\gamma \sim 1$
(see, e.g., Fig. 6 in Gloeckler and Hamilton, 1987). Thus for RC ions we obtain the dependence
$w_m \propto B^2$, which corresponds to $\beta_m(L_m) = const$.
Herewith we have made a number of simplifications. Due to the energy losses of the ions, as
well as the dependence of their trajectories on the $\mu$ value, the Liouville theorem is violated (this is
especially important with a significant spatial inhomogeneity of the PS); during strong storms, the
magnetic field is weakened even at small $L$ shells; for the $E$ and $\mu$ ranges under consideration, the
spectra of $O^+$ ions in the PS deviates from the strictly power-law form. However, these factors lead
only to weakening of the theoretical dependence $\beta_m(L_m)$, i.e. to decreases parameter $\beta_m$ with
decrease in $L_m$, and, consequently, to even greater discrepancies between this model and the
experimental results.
Very strong experimental dependence shown in Fig. 2c can be understand only if we take into
account strong variations of the fluxes and energy density of ions in the near-Earth PS and in the
region of a geosynchronous orbit during the main phase of strong storms (see, e.g., Jordanova et
al., 2010). It can be assumed that a deeper penetration of hot plasma into a geomagnetic trap is
supported not only by a stronger convection electric field, but is also provided by hot plasma with
a higher energy density in the source. A large preliminary accumulation and acceleration of ions in
the PS, especially $O^+$ ions, is apparently very important for the development of the RC on the main
phase of strong storms. Such conclusion was made earlier from the CRRES data, which was
compared with the results of mathematical modeling of the RC (see, e.g., Kozyra et al., 1998,
2002; Ebihara and Ejiri, 2003).
Strong variations in the energy density of ions in the near-Earth PS are apparently the main
reason for the large scatter of the points in Fig. 2 (this factor can also make a significant
contribution to the scatter of the points in other figures).
The distributions of the RC parameter $\beta_m$ in the $\{\beta_m, L_m\}$ space are very different for $H^+$ and $O^+$
ions, although the drift trajectories of these ions (with the same $\mu$ value) from their source in the
PS to the observation point in the RC are identical. Such difference for $H^+$ and $O^+$ ions can be
mainly associated with more significant increases of ions $O^+$ concentration (compare with $H^+$ ions)
in the PS during preliminary and main phases of the strong storms.
Thus, in the advancement of the RC towards the Earth during the main phase of storms, $O^+$ ions
play the role of the avant-guard. Protons share this role with $O^+$ ions only in the near-midnight
MLT sector; this result can be associated with more significant losses of low-energy protons
during their convection compared with $O^+$ ions (see, e.g., Kozyra et al., 1998; Kistler and Mouikis,

593 2016).

## 4.2 Ring current region at $L > L_m$

Figures 4–6 indicate that the outer region of the ionic RC is asymmetric by MLT; moreover, the
dependences of the RC parameters on $|D_{st}|$, MLT, and $L_m$, for $H^+$ and $O^+$ ions, as well as for low-
energy ($E < 60$ keV) and high-energy ions are fundamentally different.
From Figs. 4a and 5a, it can be seen that for $H^+ + O^+$ ions with $E \sim 1$–$300$ keV, the more $|D_{st}|$ at
the end of the main phase of storms, the larger parameter $w_0$ and smaller parameter $L_0$, i.e., the ion
RC increases and its outer part becomes steeper.
With that, Fig. 4a show that at the end of the main phase of storms, the contribution of ions with
$E < 60$ keV to the RC energy density decreases with an increase in $|D_{st}|$, while the contribution of
higher-energy ions systematically increases. This result is reflected also in Fig. 4c. Such effect can





be associated with an increase in the role of radial diffusion of ions to the Earth as the strength of
the storm and the duration of its main phase increases.
Figures 4c and 5c shows that for $H^+ + O^+$ ions with $E \sim 1$–300 keV, a decrease in parameter $L_m$ is
accompanied by a systematic increase in parameter $w_0$ and a decrease in parameter $L_0$. In these
results appear the opposition of the Earth's magnetic field to the RC penetration in the geomagnetic
trap (diamagnetism of the hot plasma) on the main phase of storms.
Figures 5a and 5c shows that for $H^+ + O^+$ ions with $E \sim 1$–300 keV, the more $|D_{st}|$ and less
parameter $L_m$ at the end of the main phase of storms, the smaller parameter $L_0$. These results may
indicate that for stronger storms, the outer magnetic tubes of the geomagnetic trap in the evening
and near-midnight sectors are more strongly extended towards the magnetotail; in this case, the
outer boundary of the trap approaches the Earth.
The asymmetry of the RC outer region by MLT is clearly seen in Figs. 5b, 6a, 6c and 6d.
It can be seen from Figs 6c and 6d that, in the range $E \sim 1$–300 keV, for $H^+$ ions parameter $L_0$
decreases with an increase in MLT from the evening to midnight sector, while for $O^+$ ions it
systematically increases. The significant scatter of the experimental points in these figures
(especially for $O^+$ ions) and the opposition of the trends for $H^+$ and $O^+$ ions should, generally
speaking, lead to an increase in the scatter of parameter $L_0$ for total energy density of these ions.
However, for $O^+$ ions the average value of parameter $L_0$ is smaller than for $H^+$ ions; due to this
reason, for the total energy density of ions, the trend of parameter $L_0$ by MLT is the same as for $O^+$
ions, and this correlation is better than for $O^+$ ions (see Fig. 5b).
With that, Fig. 6c, and Fig. 6d can be reconciled with Fig. 5b only by assuming that compared
to $H^+$ ions, $O^+$ ions are more evenly distributed over MLT (from evening to midnight sector). This
is directly indicated in Fig. 6a and 6b.
The differences in the dependences of parameters of the outer part of the RC on MLT for $H^+$
and $O^+$ ions are also determined by differences in the shape of the energy spectra and the spatial
distributions of these ions in the source, as well as by differences in their loss during drift in the
geomagnetic trap. It is necessary also to take into account the stronger compared with protons
variations in the energy density of $O^+$ ions in the near-Earth PS during main phase of the storms
(see Sect. 4.1).
In addition to ionization loss and loss by the interaction of ions with waves, in the outer region
of the RC, at $L > 5$–6, on the main phase of storms there are also loss of particles drifting around
the Earth, at the magnetopause, which are associated with the magnetosphere compression and
strong southern IMF (see, e.g., Kozyra et al., 2002; Ebihara and Ejiri, 2003; Keika et al., 2005); the
closer to the midday sector, the closer to the Earth this effect manifests itself.
On our distributions, the strongest influence of this mechanism one would expect for the point
5, which belongs to 14–17 MLT sector and was obtained at the end of the main phase of the giant
storm in February 1986. However, in Figs. 4a, 4c, and 5, as in Figs. 1a, 2a, and 2b, the deviations
of this point from the general trends shown in these figures by thin lines are not very large; the
point 5 is in good agreement with the regularities presented in these figures. Probably, this is
explained by the fact that the point 5 belongs to the inner region of the trap ($L = 3$–5); the radial
profile $w(L)$ at $L > 5$ was much steeper than at $L = 3$–5 (see Fig. 7 in Hamilton et al., 1988).
Note also that for penetration of the RC deep into the trap during the main phase of very strong
storms with a long main phase, its asymmetry by MLT near the RC maximum can be much smaller
than for weaker storms. This hypothesis is supported by ground-based data on storm variations in
the geomagnetic field at equatorial latitudes (see, e.g., Li et al., 2011). This effect can be related to
the fact that the radial diffusion of particles to the Earth under the action of fluctuations in the
electric and magnetic fields, which leads to a betatron acceleration of ions, proceeds faster and
more efficiently on the main phase of strong storms than during weaker storms. On the main phase





of very strong storms, the RC ions, drifting towards the Earth with the conservation of $\mu$ and $K$,
can reach lower $L$ and much higher energies, at which a significant part of these ions gets out of
control of convection, and the magnetic drift around the Earth becomes dominant for them
(symmetrical part of the RC).

## 5 Conclusions

According to the results of measurements near the plane of the geomagnetic equator from the
satellites Explorer 45, AMPTE/CCE and Van Allen Probes (A and B), on the main phase of eleven
magnetic storms of different strengths during the period from 1971 to 2016, it was made a
systematic analysis of the spatial-energy distributions of the main ionic components ($H^+$ and $O^+$) of
the ring current (RC). It is shown that behind the RC maximum, at $L > L_m$, the shape of the radial
profiles of the ions energy density of the RC is well described by the function $w(L) =$
$w_0 \exp(-L/L_0)$; parameters $w_0$ and $L_0$, characterizing the intensity and the steepness of these profiles
on $L > L_m$, have been calculated.
It has been established that the stronger the storm, the lower the average value of parameter $L_m$
of the ionic RC; however, this dependence is rather weak: $L_m \propto |D_{st}|^{-0.12}$ for ions with $E \sim 1–300$
кэВ, and $L_m \propto |D_{st}|^{-0.16}$ for ions with $E \sim 1–60$ кэВ. For ions with $E \sim 1–60$ keV, parameter $L_m$ is
smaller than for ions with $E \sim 1–300$ keV. A simple conceptual model of convection of the RC
ions on the storms main phase is considered. This model explains the experimental dependence
$L_m(|D_{st}|)$, and also the fact that in the evening and near-midnight sectors parameter $L_m$ is practically
independent from MLT.
The ratios of the energy densities of ions and the magnetic field at the RC maximum ($\beta_m$) are
calculated and it is found that for $H^+ + O^+$ ions with $E \sim 1–300$ keV the average value of
$\beta_m \propto |D_{st}|^{1.5}$. For $H^+$ ions, parameter $\beta_m$ depends by MLT and reaches its maximum values in the
pre-midnight sector. These results shows that the RC parameters depend not only on the electric
and magnetic fields and their variations, but also on the shape of the energy spectra and spatial
distributions of ions in their source, as well as the loss rates of ions during their convection.
For $O^+$ ions with $E \sim 1–300$ keV, parameter $\beta_m$ increases with a decrease in $L_m$ as $L_m^{-9.2}$. This
result shows that a deeper penetration of hot plasma into a geomagnetic trap requires not only a
stronger electric field of convection, but also a significant preliminary accumulation and
acceleration of ions (especially $O^+$ ions) in the plasma sheet (PS) of the magnetotail.
As well as, the strong dependence $\beta_m(L_m)$ for $O^+$ ions at the RC maximum, the results related to
the RC region at $L > L_m$ correspond to more significant increases of the energy density of $O^+$ ions
in the PS, compare with ions $H^+$, on the main phase of storms.
During the main phase of storms, the RC region on $L > L_m$ is asymmetric by MLT and its
parameters ($w_0$ and $L_0$) for ions of low ($E < 60$ keV) and higher energies, as well as for $H^+$ and $O^+$
ions, have different dependencies from $|D_{st}|$, MLT and $L_m$.
A strong azimuthal asymmetry of the RC ions with $E \sim 1–300$ keV on $L > L_m$ is revealed: at the
end of the main phase of storms, with an increase MLT from the evening to midnight sector,
parameter $L_0$ for ions $H^+ + O^+$ and $O^+$ systematically increases; however, for $H^+$ ions parameter $L_0$
decreases. It is found that the contribution of $O^+$ ions to the total energy density of the RC ions is
more uniformly distributed over MLT compared with the contribution of $H^+$ ions, which decreases
significantly from the midnight to the evening sector.
It is shown that for $H^+ + O^+$ ions with $E \sim 1–300$ keV, the more $|D_{st}|$ at the end of the main phase
of storms, the larger parameter $w_0$ ($w_0 \propto |D_{st}|^{1.46}$) and less parameter $L_0$ ($L_0 \propto |D_{st}|^{-0.47}$), i.e. the
outer region of the RC is enhanced and becomes steeper; with that, parameter $L_0$ is the smaller, the
nearer come to the Earth the RC: $L_0 \propto L_m^{2.6}$.



At the same time, parameter $w_0$ depends on $|D_{st}|$ and $L_m$ for $H^+$+$O^+$ ions of different energies in
different ways: at the end of the main phase of storms for ions with $E \sim$ 1–300 keV parameter $w_0$
correlates with $|D_{st}|$ ($w_0 \propto |D_{st}|^{1.46}$) and anticorrelates with $L_m$ ($w_0 \propto L_m^{-12.2}$), and for ions with $E \sim$
1–60 keV we have inverse relationships ($w_0 \propto |D_{st}|^{-1.62}$ and $w_0 \propto L_m^{11.2}$). Thus, the stronger storm
and the smaller $L_m$, the smaller fraction of low-energy ions and larger fraction of more energetic
ions in the total energy density of the RC (average kinetic energy of ions increases). Such effect
can be associated with an increase in the role of radial diffusion of ions to the Earth as the strength
of the storm and the duration of its main phase increases.
These results show also the opposition of the Earth's magnetic field to the propagation of the
RC on small $L$ (diamagnetism of the hot RC plasma) and the stretching of the geomagnetic field
towards the magnetotail on high $L$ during the main phase of storms. They reflect also differences in
the loss of ions $H^+$ and $O^+$ during their drift in the geomagnetic trap and decreases in loss with an
increase in ion energy.
*Data availability*. All data from this investigation are presented in Figs. 1–6.
*Competing interests*. The author declares that there is no conflict of interest.
*Acknowledgements*. The author thanks the Kyoto World Data Center for Geomagnetism for
providing the Dst indices.

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
