# Peer review of "Ion's ring current: regularities of the energy density distributions on the main phase of geomagnetic storms"

_Annales Geophysicae, 2023_

## Author Comment (AC1)

https://doi.org/10.5194/angeo-2023-10

***Reply to Interactive comment* by Anonymous Referee #1 from 15 May 2023 on the manuscript "Ion's ring current: regularities of the energy density distributions on the main phase of geomagnetic storms"** *by* **Alexander S. Kovtyukh**

Deeply respected Referee #1,

I am very grateful to you for such an exclusively thorough review. All these comments are very helpful for me and it is taken into account in the manuscript. The work is large, complex, but most of its intermediate results remain in my archive, and I understand and accept your doubts and concerns about the reliability of the final results presented here.

With grand regard,
Alexander S. Kovtyukh

RC1: The paper presents a study of inner magnetospheric hot ion energy density during storms, searching for patterns as a function of radial distance and local time for different ion energy ranges. The values used are based on other published studies, greatly limiting the robustness of the data set considered. The methodology is flawed. The conclusions are not particular new or significant. I find that the study is not ready for publication.

AC: I have always believed that published data that is cross-checked and well understood is the most reliable. If there are errors there, they will definitely appear in other publications of experimental data (I have always made such a comparison according to the data of different experiments).

Initially, this work did not set the task of searching for some completely new, radically different from the already known regularities in the storm dynamics of the ionic ring current (RC). The main goal here was to compare the well-known experimental results for RC ions and obtain the most harmonious overall picture. And this is not such a simple task, given that all storms are different and the RC parameters depend on the state and prehistory of the magnetosphere and heliosphere, as well as on the energy and ionic composition of the RC.

However, there are two conclusions here, which were put forward as hypotheses in the works on mathematical modeling of the RC. These hypotheses receive here quantitative experimental confirmation.

1. The conclusion that for $O^+$ ions with $E \sim 1$–$300$ keV the parameter $\beta_m \propto L_m^{-6}$ (Fig. 2c); this result shows that a deeper penetration of hot plasma into a geomagnetic trap during strong storms requires not only a stronger convection electric field, but also a significant preliminary accumulation and acceleration of ions (especially $O^+$ ions) in the RC source.

2. For the first time, it was also shown from experimental data (Fig. 4) that the greater $|D_{st}|$ at the end of the main phase of storms, the smaller the contribution of ions with $E < 60$ keV and the greater the contribution of higher-energy ions to the RC energy density (the average ion energy increases).

In addition, all other conclusions receive quantitative confirmation and analytical expression.

RC1: The introduction has no references to prior work except in two place, one where 10 "reviews" are cited all at once and other where the author's own prior work (from 2010) is cited. This method of citation is inadequate for justifying and motivating a new study. In fact, this new paper is very similar to their 2010 study, except with slightly different prior publications included.

AC: Hundreds of original experimental papers on the RC ions have been published over 50–60 years (including dozens of papers based on data from satellites of the Molniya, Horizont, and Kosmos series with the participation of the author, but these results are not suitable for solving the problems which are considered here, and I do not refer to them). Many hundreds of works on mathematical modeling of the RC have been performed also.

Since the problems considered here concerns almost all aspects of the RC dynamics, it is impossible to give a sufficiently complete list of these works here, and I limited myself only to the most complete reviews (not including my reviews on the RC in monographs published in Russia), and original works that are directly related to the set tasks here (they are given in the following text and in the References).

Kovtyukh (2010) was considered only the position of the RC energy density maximum and the parameter $\beta_m$ at this maximum depending on the value of $|D_{st}|$, as well as the *shape* of the RC outer edge. In this work only the energy densities of protons and the total energy densities of all RC ions were considered; separate analysis for protons and oxygen ions, as well as for ion energy and MLT was not carried out; the dependences of the parameters of the outer edge of the RC on $|D_{st}|$ and MLT were not considered. In 2010, there was also no data from the Van Allen Probes satellites, which play a very important role here.

It would be possible to bring the Introduction to the normal generally accepted form if I knew other articles with such a statement of the problem. And I very carefully read all publications on RC ions, especially experimental ones, over the past 50 years.

RC1: It only uses 11 storm events, some with repeated entries, to increase the number of entries in Table 1 up to 17. I don't understand the selection of events and timings for this table, nor the focus on these events. This is not a systematic study of hot ion energy content, but a highly skewed listing based on the author's selection of previously published studies. There is no way to verify the robustness of the results.

AC: The classification and selection of the experimental data considered here are described in detail in Section 2.

In Table. 1, as in other tables, there are no duplicate entries; they all refer to different UT, and if they are at close UT, then they are obtained by different instruments (this applies only to data from the Van Allen Probes satellites).

The choice of experimental results, events, and time is not accidental here and covers almost all results on the radial energy density profiles of $H^+$ and $O^+$ RC ions near the equatorial plane, published from 1973 to 2022 (not counting repeated publications).

Many other publications on RC ions can be added to this, but they were obtained in regions that deviate significantly from the equatorial plane (on the OGO-3, Molniya-1, Polar satellites, etc.), as well as on low-altitude polar satellites. Geostationary orbit is also not suitable for our purposes.

In addition, in the last 10-20 years, most of results on the RC are presented only in the form of color spectrograms of ion fluxes. Such spectrograms well illustrate the evolution of the fluxes and spectra of ions in the satellite orbit, but from them it is impossible to extract sufficiently accurate flux values (and to determine the shape of the spectra), which is necessary for calculating the ion energy density and constructing the radial profiles of the RC.

To check the reliability of the results presented here, it is enough to open the corresponding article indicated in the last column of Table. 1.

RC1: Conducting line fits with fewer than 10 points makes the resulting fit highly susceptible to outliers. It also takes a very high R to reach the traditional p=0.05 level of statistical significance. For example, neither of the R values listed on page 5 (end of section 3.1) reach this p value. That is, even with these seemingly high correlation coefficients, the fit has a decent probability of arising from random chance. In any case, a linear fit of 3 points (the second equation) is almost never a reasonable scientific method.

AC: I agree. Text will be corrected.

RC1: I have a side comment on visualization in this paper. I greatly dislike the inclusion of points that are then not used in the fits (Figures 1, 2, 5, and 6), or the inclusion of two line fits from subsets of the points on the same graph (Figure 4). This obscures the true connection between points and linear fits. It is fine to show them all together, but then also make a separate plot to show what points go with what fit.

AC: Figure 1 shows all 17 points corresponding to the Table 1.

The physical reasons why from Figs. 2, 4, 5, and 6, the certain experimental points are excluded or not taken into account in the calculations of correlation dependencies, in each case (for each figure) are specified in the text of the article.

The lines in Fig. 4 very clearly show mutually opposite tendencies in these dependences for ions of high and low energies, and it is undesirable to separate them.

RC1: For Table 2, the peak energy density column has values given to one or two significant figures. The beta values are then given to 3 significant figures, which is unjustified. They should all be reported only to one significant digit. The uncertainty on all of the beta values is, therefore, large. This most liinvalidates the linear fits found from

the plots of beta versus 3 parameters in Figure 2. While the R values of the two fits reach the p=0.05 level, this extra uncertainty makes them not meaningful.

> AC: I agree. Table 2 (lines 3, 6, 9, 14, 15) and Fig. 2 are corrected.

> But these changes do not exceed a few percent, and practically they do not affect the correlation curves in Fig. 2.

RC1: The analysis at the beginning of section 3.3, built around Table 3 and Figure 3, appear to be based on exponential curve parameter values from only two L value energy densities. This chosen functional form is not justified with only two points. This analysis does not defend the "well described" word choice in the text.

> AC: Considering Fig. 3, one can indeed assume that the curves shown on it are built along two extreme points, but the text of the article explains how this figure was built. In Appendix 1, these curves are given along with the experimental points (these figures take up too much space and therefore were excluded from the article and replaced by the general Fig. 3).

RC1: I have the same complaints about Figures 4, 5, and 6 as I do about Figures 1 and 2. There are too few points, chosen by an unsystematic selection of storm events, to make the results meaningful.

> AC: The choice of the experimental points is by no an unsystematic (see above).

RC1: I think that the issues with the methodology of section 3 invalidate any conclusions drawn from the discussion in section 4.

> AC: I hope that the discussion held here removes this objection.

RC1: The conclusions drawn are not new. That the ions should move closer to the Earth during larger storm events is well known. A maximum energy density in the pre-midnight sector is expected based on drift physics. The same can be said about the energy dependence of MLT features.

> AC: It has long been known that with increasing geomagnetic activity, the RC approaches the Earth. But how strong is this dependence?

> Many specific questions also arise regarding the asymmetry of the RC, although in general it is understandable and has long been known.

> These and other questions on the RC require further comprehensive analysis. Different mathematical models give different answers to these and many other questions for the RC.

> In this work, two new conclusions were obtained, which were previously put forward as imprecise hypotheses, and here they receive experimental confirmation and a quantitative expression (see above).

RC1: The only suggestion that I can make to raise the robustness: download the data for many more storms and calculate these dependences with a systematic approach to all possible observations. Instead of relying on other studies to identify storms and do the initial analysis, calculate the energy densities directly from the data.

AC: Unfortunately, the problems considered here cannot be solved in such a simple way. The problem of the RC is very complex, and by purely statistical studies (they are also very important), it can hardly be solved completely.

It would be possible to take as a basis the experimental data on the ion fluxes of the RC, presented in the Internet, and carry out a more complete statistical analysis of the parameters of the RC for several dozen storms, but sufficiently complete data on the RC suitable for the problems considered here are not available in the Internet (they are not freely available).

In addition, such data may contain large methodological errors associated with the transmission of information, with noise and illumination of devices, background fluxes, and problems of ion selection by charge and mass.

Kind regards,
Alexander Kovtyukh

---

## Author Comment (AC3)

| | $L$ | $\Delta L$ | $(\Delta L)^2$ | $\Delta x \Delta y$ | $W$ | $\ln W$ | $\Delta \ln W$ | $(\Delta \ln W)^2$ |
|---|---|---|---|---|---|---|---|---|
| | 3.5 | -0.75 | 0.5625 | -0.2811 | 46 | 3.8286 | 0.3748 | 0.1405 |
| | 4 | -0.25 | 0.0625 | -0.0254 | 35 | 3.5553 | 0.1015 | 0.0103 |
| | 4.5 | 0.25 | 0.0625 | -0.0395 | 27 | 3.2958 | -0.1580 | 0.0250 |
| | 5 | .0.75 | 0.5625 | -0.2387 | 23 | 3.1355 | -0.3183 | 0.1013 |
| $\sum$ | 17 | 0 | 1.25 | -0.5847 | | 13.8152 | 0 | 0.2771 |
| $\langle\rangle$ | 4.25 | 0 | 0.3125 | -0.1462 | | 3.4538 | 0 | 0.0693 |
| $s_x^2$ | | | 0.4167 | | | | | |
| $s_x$ | | | 0.6455 | | | | | |
| $s_y^2$ | | | | | | | | 0.09237 |
| $s_y$ | | | | | | | | 0.30392 |
| $s_{xy}$ | | | | -0.1949 | | | | |

$R = -\,0.993$

$W = W_0\, e^{-L/L0}$

$\ln W = \ln W_0 - L/L_0 = 5.4416 - L/2.138$

$W = 230\, \exp(-L/2.138)$

| | $L$ | $\Delta L$ | $(\Delta L)^2$ | $\Delta x \Delta y$ | $W$ | $\ln W$ | $\Delta \ln W$ | $(\Delta \ln W)^2$ |
|---|---|---|---|---|---|---|---|---|
| | 3.5 | -0.75 | 0.5625 | -0.1894 | 20 | 2.996 | 0.2525 | 0.0638 |
| | 4 | -0.25 | 0.0625 | -0.0069 | 18 | 2.890 | 0.1465 | 0.0215 |
| | 4.5 | 0.25 | 0.0625 | -0.0351 | 13.5 | 2.603 | -0.1405 | 0.0197 |
| | 5 | .0.75 | 0.5625 | -0.1939 | 12 | 2.485 | -0.2585 | 0.0668 |
| $\sum$ | 17 | 0 | 1.25 | -0.4253 | | 10.974 | 0 | 0.1718 |
| $\langle\rangle$ | 4.25 | 0 | 0.3125 | -0.1063 | | 2.7435 | 0 | 0.043 |
| $s_x^2$ | | | 0.4167 | | | | | |
| $s_x$ | | | 0.6455 | | | | | |
| $s_y^2$ | | | | | | | | 0.05727 |
| $s_y$ | | | | | | | | 0.2393 |
| $s_{xy}$ | | | | -0.1418 | | | | |

$R = -\,0.918$

$W = W_0\, e^{-L/L0}$

$\ln W = \ln W_0 - L/L_0 = 4.19 - L/2.938$

$W = 66\, \exp(-L/2.94)$

| | $L$ | $\Delta L$ | $(\Delta L)^2$ | $\Delta x \Delta y$ | $W$ | $\ln W$ | $\Delta \ln W$ | $(\Delta \ln W)^2$ |
|---|---|---|---|---|---|---|---|---|
| | 4.5 | -1.5 | 2.25 | -0.9440 | 9.7 | 2.272 | 0.6293 | 0.3960 |
| | 5 | -1 | 1 | -0.4493 | 8.1 | 2.092 | 0.4493 | 0.2019 |
| | 5.5 | -0.5 | 0.25 | -0.1447 | 6.9 | 1.932 | 0.2893 | 0.0837 |
| | 6 | 0 | 0 | 0 | 5.2 | 1.649 | 0.0063 | 0.0000 |
| | 6.5 | 0.5 | 0.25 | -0.1674 | 3.7 | 1.308 | -0.3347 | 0.1120 |
| | 7 | 1 | 1 | -0.3897 | 3.5 | 1.253 | -0.3897 | 0.1519 |
| | 7.5 | 1.5 | 2.25 | -0.9746 | 2.7 | 0.993 | -0.6497 | 0.4221 |
| $\sum$ | 42 | 0 | 7 | -3.0697 | | 11.499 | 0.0001 | 1.3676 |
| $\langle \rangle$ | 6 | 0 | 1 | -0.4385 | | 1.6427 | 0 | 0.1954 |
| $s_x^2$ | | | 1.1667 | | | | | |
| $s_x$ | | | 1.0801 | | | | | |
| $s_y^2$ | | | | | | | | 0.2279 |
| $s_y$ | | | | | | | | 0.4774 |
| $s_{xy}$ | | | | -0.5116 | | | | |

$R = -0.992$

$W = W_0 \, e^{-L/L0}$

$\ln W = \ln W_0 - L/L_0 = 4.2737 - L/2.28$

$W = 72 \exp(-L/2.28)$

| | $L$ | $\Delta L$ | $(\Delta L)^2$ | $\Delta x \Delta y$ | $W$ | $\ln W$ | $\Delta \ln W$ | $(\Delta \ln W)^2$ |
|---|---|---|---|---|---|---|---|---|
| | 3.5 | -1.5 | 2.25 | -1.425 | 42 | 3.738 | 0.95 | 0.9025 |
| | 4 | -1 | 1 | -0.767 | 35 | 3.555 | 0.767 | 0.5883 |
| | 4.5 | -0.5 | 0.25 | -0.195 | 24 | 3.178 | 0.39 | 0.1521 |
| | 5 | 0 | 0 | 0 | 16 | 2.773 | -0.015 | 0.0002 |
| | 5.5 | 0.5 | 0.25 | -0.195 | 11 | 2.398 | -0.39 | 0.1521 |
| | 6 | 1 | 1 | -0.709 | 8 | 2.079 | -0.709 | 0.5927 |
| | 6.5 | 1.5 | 2.25 | -1.494 | 6 | 1.792 | -0.996 | 0.9920 |
| $\sum$ | 35 | 0 | 7 | -4.785 | | 19.513 | -0.003 | 3.3799 |
| $\langle \rangle$ | 5 | 0 | 1 | -0.6836 | | 2.788 | 0 | 0.4828 |
| $s_x^2$ | | | 1.167 | | | | | |
| $s_x$ | | | 1.0801 | | | | | |
| $s_y^2$ | | | | | | | | 0.5633 |
| $s_y$ | | | | | | | | 0.7505 |
| $s_{xy}$ | | | | -0.7975 | | | | |

$R = -0.984$

$W = W_0 \, e^{-L/L0}$

$\ln W = \ln W_0 - L/L_0 = 6.2061 - L/1.463$

$W = 496 \exp(-L/1.463)$

|  | $L$ | $\Delta L$ | $(\Delta L)^2$ | $\Delta x \Delta y$ | $W$ | $\ln W$ | $\Delta \ln W$ | $(\Delta \ln W)^2$ |
|---|---|---|---|---|---|---|---|---|
|  | 3 | -1 | 1 | -0.925 | 80 | 4.382 | 0.925 | 0.856 |
|  | 3.5 | -0.5 | 0.25 | -0.210 | 48.3 | 3.877 | 0.420 | 0.176 |
|  | 4 | 0 | 0 | 0 | 40 | 3.695 | 0.238 | 0.057 |
|  | 4.5 | 0.5 | 0.25 | -0.171 | 16 | 2.773 | -0.684 | 0.468 |
|  | 5 | 1 | 1 | -0.901 | 13 | 2.556 | -0.901 | 0.812 |
| $\sum$ | 20 | 0 | 2.5 | -2.207 |  | 17.283 | -0.002 | 2.399 |
| $\langle\rangle$ | 4 | 0 | 0.5 | -0.441 |  | 3.457 | 0 | 0.480 |
| $s_x^2$ |  |  | 0.625 |  |  |  |  |  |
| $s_x$ |  |  | 0.791 |  |  |  |  |  |
| $s_y^2$ |  |  |  |  |  |  |  | 0.600 |
| $s_y$ |  |  |  |  |  |  |  | 0.774 |
| $s_{xy}$ |  |  |  | -0.552 |  |  |  |  |

$R = -0.901$

$W = W_0\, e^{-L/L0}$

$\ln W = \ln W_0 - L/L_0 = 6.984 - L/1.134$

$W = 1079 \exp(-L/1.134)$

|  | $L$ | $\Delta L$ | $(\Delta L)^2$ | $\Delta x \Delta y$ | $W$ | $\ln W$ | $\Delta \ln W$ | $(\Delta \ln W)^2$ |
|---|---|---|---|---|---|---|---|---|
|  | 3.5 | -0.75 | 0.5625 | -0.330 | 47.8 | 3.867 | 0.440 | 0.1936 |
|  | 4 | -0.25 | 0.0625 | -0.067 | 40.25 | 3.695 | 0.268 | 0.0718 |
|  | 4.5 | 0.25 | 0.0625 | -0.015 | 29 | 3.367 | -0.060 | 0.0036 |
|  | 5 | 0.75 | 0.5625 | -0.486 | 16.1 | 2.779 | -0.648 | 0.4199 |
| $\sum$ | 17 | 0 | 1.25 | -0.898 |  | 13.708 | 0 | 0.6889 |
| $\langle\rangle$ | 4.25 | 0 | 0.3125 | -0.2245 |  | 3.427 | 0 | 0.1722 |
| $s_x^2$ |  |  | 0.4167 |  |  |  |  |  |
| $s_x$ |  |  | 0.6455 |  |  |  |  |  |
| $s_y^2$ |  |  |  |  |  |  |  | 0.2296 |
| $s_y$ |  |  |  |  |  |  |  | 0.4792 |
| $s_{xy}$ |  |  |  | -0.2993 |  |  |  |  |

$R = -0.968$

$W = W_0\, e^{-L/L0}$

$\ln W = \ln W_0 - L/L_0 = 6.47996 - L/1.3921$

$W = 652 \exp(-L/1.392)$

| 10 | L | ΔL | (ΔL)² | ΔxΔy | W | ln W | Δln W | (Δln W)² |
|---|---|---|---|---|---|---|---|---|
| | 3.5 | -1 | 1 | -0.791 | 28 | 3.332 | 0.791 | 0.626 |
| | 4 | -0.5 | 0.25 | -0.252 | 21 | 3.045 | 0.504 | 0.254 |
| | 4.5 | 0 | 0 | 0 | 14 | 2.639 | 0.098 | 0.010 |
| | 5 | 0.5 | 0.25 | -0.231 | 8 | 2.079 | -0.462 | 0.213 |
| | 5.5 | 1 | 1 | -0.932 | 5 | 1.609 | -0.932 | 0.869 |
| Σ | 22.5 | 0 | 2.5 | -2.206 | | 12.704 | -0.001 | 1.972 |
| ⟨⟩ | 4.5 | 0 | 0.5 | 0.441 | | 2.541 | 0 | 0.394 |
| $s_x^2$ | | | 0.625 | | | | | |
| $s_x$ | | | 0.791 | | | | | |
| $s_y^2$ | | | | | | | | 0.493 |
| $s_y$ | | | | | | | | 0.702 |
| $s_{xy}$ | | | | -0.5515 | | | | |

$R = -0.999$

$W = W_0\, e^{-L/L0}$

$\ln W = \ln W_0 - L/L_0 = 6.512 - L/1.133$

$W = 673 \exp(-L/1.133)$

[Figure]

| 10 | L | ΔL | (ΔL)² | ΔxΔy | W | ln W | Δln W | (Δln W)² |
|---|---|---|---|---|---|---|---|---|
| | 3.5 | -1 | 1 | -0.734 | 32 | 3.466 | 0.734 | 0.539 |
| | 4 | -0.5 | 0.25 | -0.2015 | 23 | 3.135 | 0.403 | 0.162 |
| | 4.5 | 0 | 0 | 0 | 15.5 | 2.741 | 0.009 | 0.0001 |
| | 5 | 0.5 | 0.25 | -0.2145 | 10 | 2.303 | -0.429 | 0.184 |
| | 5.5 | 1 | 1 | -0.717 | 7.5 | 2.015 | -0.717 | 0.514 |
| Σ | 22.5 | 0 | 2.5 | -1.867 | | 13.66 | 0.000 | 1.399 |
| ⟨⟩ | 4.5 | 0 | 0.5 | -0.3734 | | 2.732 | 0.000 | 0.28 |
| $s_x^2$ | | | 0.625 | | | | | |
| $s_x$ | | | 0.791 | | | | | |
| $s_y^2$ | | | | | | | | 0.35 |
| $s_y$ | | | | | | | | 0.5914 |
| $s_{xy}$ | | | | -0.467 | | | | |

$R = -0.998$

$W = W_0\, e^{-L/L0}$

$\ln W = \ln W_0 - L/L_0 = 6.0944 - L/1.34$

$W = 443 \exp(-L/1.34)$

| 10 | $L$ | $\Delta L$ | $(\Delta L)^2$ | $\Delta x\Delta y$ | $W$ | $\ln W$ | $\Delta\ln W$ | $(\Delta\ln W)^2$ |
|---|---|---|---|---|---|---|---|---|
| | 3.5 | -0.5 | 0.25 | -0.2275 | 12.5 | 2.526 | 0.455 | 0.2070 |
| | 4 | 0 | 0 | 0 | 8 | 2.079 | 0.008 | 0.0000 |
| | 4.5 | 0.5 | 0.25 | -0.231 | 5 | 1.609 | -0.462 | 0.2134 |
| $\Sigma$ | 12 | 0 | 0.5 | -0.4585 | | 6.214 | 0.001 | 0.4204 |
| $\langle\rangle$ | 4 | 0 | 0.167 | -0.1528 | | 2.071 | 0 | 0.1401 |
| $s_x^2$ | | | 0.25 | | | | | |
| $s_x$ | | | 0.5 | | | | | |
| $s_y^2$ | | | | | | | | 0.2102 |
| $s_y$ | | | | | | | | 0.4585 |
| $s_{xy}$ | | | | -0.2293 | | | | |

$R = -0.999$

$W = W_0\, e^{-L/L0}$

$\ln W = \ln W_0 - L/L_0 = 5.74 - L/1.09$

$W = 311\exp(-L/1.09)$

| | $L$ | $\Delta L$ | $(\Delta L)^2$ | $\Delta x\Delta y$ | $W$ | $\ln W$ | $\Delta\ln W$ | $(\Delta\lg W)^2$ |
|---|---|---|---|---|---|---|---|---|
| | 3.5 | -1 | 1 | -0.530 | 16 | 2.773 | 0.530 | 0.2809 |
| | 4 | -0.5 | 0.25 | -0.142 | 12.5 | 2.526 | 0.283 | 0.0801 |
| | 4.5 | 0 | 0 | 0.000 | 9 | 2.197 | -0.046 | 0.0021 |
| | 5 | 0.5 | 0.25 | -0.114 | 7.5 | 2.015 | -0.228 | 0.0520 |
| | 5.5 | 1 | 1 | -0.538 | 5.5 | 1.705 | -0.538 | 0.2894 |
| $\Sigma$ | 22.5 | 0 | 2.5 | -1.324 | | 11.216 | 0.001 | 0.7045 |
| $\langle\rangle$ | 4.5 | 0 | 0.5 | -0.2648 | | 2.243 | 0 | 0.1409 |
| $s_x^2$ | | | 0.625 | | | | | |
| $s_x$ | | | 0.791 | | | | | |
| $s_y^2$ | | | | | | | | 0.1761 |
| $s_y$ | | | | | | | | 0.4197 |
| $s_{xy}$ | | | | -0.331 | | | | |

$R = -0.997$

$W = W_0\, e^{-L/L0}$

$\ln W = \ln W_0 - L/L_0 = 4.626 - L/1.89$

$W = 102\exp(-L/1.89)$

| | $L$ | $\Delta L$ | $(\Delta L)^2$ | $\Delta x \Delta y$ | $W$ | $\ln W$ | $\Delta \ln W$ | $(\Delta \ln W)^2$ |
|---|---|---|---|---|---|---|---|---|
| | 3 | -1 | 1 | -0.689 | 28 | 3.332 | 0.689 | 0.4747 |
| | 3.5 | -0.5 | 0.25 | -0.1095 | 17.5 | 2.862 | 0.219 | 0.0480 |
| | 4 | 0 | 0 | 0 | 15 | 2.708 | 0.065 | 0.0042 |
| | 4.5 | 0.5 | 0.25 | -0.1005 | 11.5 | 2.442 | -0.201 | 0.0404 |
| | 5 | 1 | 1 | -0.771 | 6.5 | 1.872 | -0.771 | 0.5944 |
| $\sum$ | 20 | 0 | 2.5 | -1.67 | | 13.216 | 0.001 | 1.1617 |
| $\langle \rangle$ | 4 | 0 | 0.5 | -0.334 | | 2.643 | 0 | 0.2323 |
| $s_x^2$ | | | 0.625 | | | | | |
| $s_x$ | | | 0.791 | | | | | |
| $s_y^2$ | | | | | | | | 0.29 |
| $s_y$ | | | | | | | | 0.5389 |
| $s_{xy}$ | | | | -0.4175 | | | | |

$R = -0.979$

$W = W_0\, e^{-L/L0}$

$\ln W = \ln W_0 - L/L_0 = 5.315 - L/1.497$

$W = 203\, \exp(-L/1.497)$

| | $L$ | $\Delta L$ | $(\Delta L)^2$ | $\Delta x \Delta y$ | $W$ | $\ln W$ | $\Delta \ln W$ | $(\Delta \ln W)^2$ |
|---|---|---|---|---|---|---|---|---|
| | 3.5 | -1 | 1 | -0.564 | 55 | 4.007 | 0.564 | 0.3181 |
| | 4 | -0.5 | 0.25 | -0.148 | 42 | 3.738 | 0.295 | 0.0870 |
| | 4.5 | 0 | 0 | 0 | 31 | 3.434 | -0.009 | 0.0001 |
| | 5 | 0.5 | 0.25 | -0.192 | 21.3 | 3.059 | -0.384 | 0.1475 |
| | 5.5 | 1 | 1 | -0.468 | 19.6 | 2.975 | -0.468 | 0.2190 |
| $\sum$ | 22.5 | 0 | 2.5 | -1.372 | | 17.213 | -0.002 | 0.7717 |
| $\langle \rangle$ | 4.5 | 0 | 0.5 | -0.2744 | | 3.443 | 0 | 0.1543 |
| $s_x^2$ | | | 0.625 | | | | | |
| $s_x$ | | | 0.791 | | | | | |
| $s_y^2$ | | | | | | | | 0.193 |
| $s_y$ | | | | | | | | 0.439 |
| $s_{xy}$ | | | | -0.343 | | | | |

$R = -0.988$

$W = W_0\, e^{-L/L0}$

$\ln W = \ln W_0 - L/L_0 = 5.91 - L/1.824$

$W = 368\, \exp(-L/1.824)$

|  | $L$ | $\Delta L$ | $(\Delta L)^2$ | $\Delta x \Delta y$ | $W$ | $\ln W$ | $\Delta \ln W$ | $(\Delta \ln W)^2$ |
|---|---|---|---|---|---|---|---|---|
|  | 3.5 | -1 | 1 | -0.5616 | 15.30 | 2.7278 | 0.5616 | 0.3154 |
|  | 4 | -0.5 | 0.25 | -0.12795 | 11.27 | 2.4221 | 0.2559 | 0.0655 |
|  | 4.5 | 0 | 0 | 0 | 8.37 | 2.1246 | -0.0416 | 0.0017 |
|  | 5 | 0.5 | 0.25 | -0.13955 | 6.60 | 1.8871 | -0.2791 | 0.0779 |
|  | 5.5 | 1 | 1 | -0.4966 | 5.31 | 1.6696 | -0.4966 | 0.2466 |
| $\sum$ | 22.5 | 0 | 2.5 | -1.3257 |  | 10.8312 | 0.0002 | 0.7071 |
| $\langle\rangle$ | 4.5 | 0 | 0.5 | -0.2651 |  | 2.1662 | 0 | 0.1414 |
| $s_x^2$ |  |  | 0.625 |  |  |  |  |  |
| $s_x$ |  |  | 0.791 |  |  |  |  |  |
| $s_y^2$ |  |  |  |  |  |  |  | 0.1768 |
| $s_y$ |  |  |  |  |  |  |  | 0.4204 |
| $s_{xy}$ |  |  |  | -0.3314 |  |  |  |  |

$R = -0.997$

$W = W_0\, e^{-L/L0}$

$\ln W = \ln W_0 - L/L_0 = 4.5523 - L/1.886$

$W = 95 \exp(-L/1.886)$

|  | $L$ | $\Delta L$ | $(\Delta L)^2$ | $\Delta x \Delta y$ | $W$ | $\ln W$ | $\Delta \ln W$ | $(\Delta \ln W)^2$ |
|---|---|---|---|---|---|---|---|---|
|  | 3.5 | -0.75 | 0.5625 | -0.3465 | 24.47 | 3.197 | 0.462 | 0.2134 |
|  | 4 | -0.25 | 0.0625 | -0.0348 | 17.71 | 2.874 | 0.139 | 0.0193 |
|  | 4.5 | 0.25 | 0.0625 | -0.0295 | 13.69 | 2.617 | -0.118 | 0.0139 |
|  | 5 | 0.75 | 0.5625 | -0.3630 | 9.50 | 2.251 | -0.484 | 0.2343 |
| $\sum$ | 17 | 0 | 1.25 | -0.7738 |  | 10.939 | -0.001 | 0.4809 |
| $\langle\rangle$ | 4.25 | 0 | 0.3125 | -0.1935 |  | 2.735 | 0 | 0.1202 |
| $s_x^2$ |  |  | 0.4167 |  |  |  |  |  |
| $s_x$ |  |  | 0.6255 |  |  |  |  |  |
| $s_y^2$ |  |  |  |  |  |  |  | 0.1603 |
| $s_y$ |  |  |  |  |  |  |  | 0.4004 |
| $s_{xy}$ |  |  |  | -0.2579 |  |  |  |  |

$R = -0.998$

$W = W_0\, e^{-L/L0}$

$\ln W = \ln W_0 - L/L_0 = 5.366 - L/1.616$

$W = 214 \exp(-L/1.616)$

| | $L$ | $\Delta L$ | $(\Delta L)^2$ | $\Delta x\Delta y$ | $W$ | $\ln W$ | $\Delta\ln W$ | $(\Delta\ln W)^2$ |
|---|---|---|---|---|---|---|---|---|
| | 3 | -1 | 1 | -0.772 | 17 | 2.833 | 0.772 | 0.5960 |
| | 3.5 | -0.5 | 0.25 | -0.1685 | 11 | 2.398 | 0.337 | 0.1136 |
| | 4 | 0 | 0 | 0 | 8 | 2.079 | 0.018 | 0.0003 |
| | 4.5 | 0.5 | 0.25 | -0.226 | 5 | 1.609 | -0.452 | 0.2043 |
| | 5 | 1 | 1 | -0.3375 | 4 | 1.386 | -0.675 | 0.4556 |
| $\sum$ | 20 | 0 | 2.5 | -1.504 | | 10.305 | 0 | 1.3698 |
| $\langle\rangle$ | 4 | 0 | 0.5 | -0.301 | | 2.061 | 0 | 0.274 |
| $s_x^2$ | | | 0.625 | | | | | |
| $s_x$ | | | 0.791 | | | | | |
| $s_y^2$ | | | | | | | | 0.3425 |
| $s_y$ | | | | | | | | 0.5852 |
| $s_{xy}$ | | | | -0.376 | | | | |

$R = -0.812$

$W = W_0\, e^{-L/L0}$

$\ln W = \ln W_0 - L/L_0 = 4.4674 - L/1.662$

$W = 87\exp(-L/1.662)$

| | $L$ | $\Delta L$ | $(\Delta L)^2$ | $\Delta x\Delta y$ | $W$ | $\ln W$ | $\Delta\ln W$ | $(\Delta\ln W)^2$ |
|---|---|---|---|---|---|---|---|---|
| | 3 | -1.5 | 2.25 | -1.782 | 70 | 4.248 | 1.188 | 1.411 |
| | 3.5 | -1 | 1 | -0.629 | 40 | 3.689 | 0.629 | 0.396 |
| | 4 | -0.5 | 0.25 | -0.032 | 20 | 2.996 | -0.064 | 0.004 |
| | 4.5 | 0 | 0 | 0 | 18 | 2.890 | -0.17 | 0.029 |
| | 5 | 0.5 | 0.25 | -0.1135 | 17 | 2.833 | -0.227 | 0.052 |
| | 5.5 | 1 | 1 | -0.494 | 13 | 2.565 | -0.494 | 0.244 |
| | 6 | 1.5 | 2.25 | -1.294 | 9 | 2.197 | -0.863 | 0.745 |
| $\sum$ | 31.5 | 0 | 7 | -4.345 | | 21.148 | -0.001 | 2.881 |
| $\langle\rangle$ | 4.5 | 0 | 1 | -0.621 | | 3.0597 | 0 | 0.412 |
| $s_x^2$ | | | 1.1666 | | | | | |
| $s_x$ | | | 1.0801 | | | | | |
| $s_y^2$ | | | | | | | | 0.4892 |
| $s_y$ | | | | | | | | 0.69294 |
| $s_{xy}$ | | | | -0.724 | | | | |

$R = -0.967$

$W = W_0\, e^{-L/L0}$

$\ln W = \ln W_0 - L/L_0 = 5.853 - L/1.611$

$W = 348\exp(-L/1.611)$

---

## Author Comment (AC4)

$W = 66 \exp(-L/2.94)$

$R = -0.918$

---

## Author Comment (AC5)

https://doi.org/10.5194/angeo-2023-10
***Interactive comment*** by Anonymous Referee #2 from 25 May 2023 on the manuscript "Ion's ring current: regularities of the energy density distributions on the main phase of geomagnetic storms" *by* Alexander S. Kovtyukh

Deeply respected Referee #2,

I am very grateful to you for an exclusively generous and thorough review. All these comments are very helpful for me and it is taken into account in the manuscript.

With grand regard,
Alexander S. Kovtyukh

GENERAL COMMENT

RC2: The author aims to determine the characteristics of the ring current, by analyzing the spatial distributions of the energy density of energetic ions during the main phase of magnetic storms. According to Table 1 of this manuscript, the data used in this study are obtained from those presented in published papers. It seems to me that the author has read the data values (Universal time, Magnetic local time, L-value, and Dst index, all of which correspond to the satellite observation time of the maximum energy density) from the figures in the papers. However, the manuscript does not adequately describe from which figure the author extracted the data values (see Specific Comments #1 below). It is difficult for me to evaluate the author's analysis and interpretations until the data values used in the manuscript are enough reliable.

AC: Yes, I agree. The numbers of the corresponding figures from the cited papers have been added to the last column of Table 1 (see Appendix AC6). In addition, Appendix AC7 contains all the main figures which were used in my work. Only Fig. 2 from Fritz et al. (1974), corresponding to Line 2 in Table 1, unfortunately have a poor quality, a gray background, low resolution, and it is slightly tilted to the left.

In addition to the figures given in Appendix AC7, full information from the corresponding papers has been used to link these figures to UT, MLT, and *L*. The corresponding $D_{st}$ index values were taken from wdc.kugi.kyoto-u.ac.jp/dst_final/index.html.

SPECIFIC COMMENTS

RC2: 1. The author should mention about the figures and/or tables that she/he used to extract all the data values, UT, MLT, Lm, and |Dst|, listed in Table 1.

I have checked two of the listed references: Yue et al, 2018 and Keika et al., 2018; and found it difficult for me to extract the data and even find some data.

For example, in the paper by Yue et al., 2018, pressure was at maximum at ~13:30 UT, which is different from UT in Table 1. The SYM-H index is presented in the paper, but the Dst index is not presented. The L value is not presented in the paper; Lm (L for the maximum energy density) is not mentioned.

In the paper by Keika et al., 2018, Lm is presented, but the corresponding UT and MLT are not presented/mentioned.

If the author obtained those data partly from the published figures/tables and partly by her/himself from the original data files provided by the mission teams, please elaborate on the processes.

AC: I have added to the Table 1 numbers of the main figures used in my work (see Appendix AC6). The corresponding figures are given in Appendix AC7. Text also will be corrected.

With data binding by Yue et al. (2018) I have been the biggest difficulties and here I was wrong. You are right: the pressure have maximum at ~13.30 UT (more precisely, at 13.10), and not at 16.30 UT. The $L$ scale is not given in this work, and it is possible to determine the RC parameters for $L > L_m$ only with large errors, but the $L_m$ value may be found in the Supporting Information for this work. The UT value in Line 13 of the Table 1 changed to 13.10, and $|D_{st}|$ value changed from 72 to 58 nT; a position of point 13 in Figs. 1a and 2a on the scale $|D_{st}|$ are also corrected. The $D_{st}$ index values were taken from wdc.kugi.kyoto-u.ac.jp/dst_final/index.html.

The work Keika et al. (2018) provides, although in a very complex form, all the necessary information. Pass A4 and B5 were used (the UT intervals for them are shown in Figs. 4 and 8 of this work), and the MLT values were calculated from the orbits of the satellites shown in Fig. 2.

RC2: 2. The early part of Introduction (Lines 30-46) and latter part (lines 66-72) have not cited any published papers, although the paragraphs contain our current understanding based on a large number of previous studies.

AC: Many hundreds of references would have to be cited here. In this section, the problem statement for this work is given, and I limited myself here only to links to the main reviews on the RC. Citations of the original works on the RC go already at the beginning of the next section and continue until the end of this manuscript.

RC2: 3. The dipole magnetic field is used to calculate w_Bd, but it is well accepted that the magnetic field configuration on the night side is significantly deviated from the dipole during a storm, particularly during the main phase. In addition, magnetic field data are available at the time of the maximum energy density (i.e., at Lm) for most of the storms listed in Table 1. The author should use either a better magnetic field model or in-situ observations.

AC: Calculations of $\beta_{md}$ values for the simplest model of a dipole magnetic field are given only in column 4 of Table 2 and are not used in Fig. 2. These values are given only for comparison with the values of $\beta_m$ given in column 5 of Table. 2.

When calculating the values of $\beta_m$ given in columns 5, 6, and 7 of Table 2, the magnetic effect of the RC is taken into account, which is the main correction for the magnetic field in the region of the RC maximum (see, e. g., Cahill, 1973; Cahill and Lee, 1975; Berko et al., 1975; Lyons, 1977; Krimigis et al., 1985; Lui et al., 1987).

In addition, for most of the storms considered here, there are current values of the magnetic field in the orbits of the corresponding satellites. These values of the magnetic field are also used here to correct the calculations of the parameters $\beta_m$, although I do not always refer to these publications (I did not want to make the bibliography larger than the main text).

RC2: 4. The author says that the ring current ion energy density is well approximated by an exponential function. It is important to present quantitatively to what extent it can be approximated.

AC: Such an approximation is illustrated by figures supplementing Fig. 3 and given in Appendixes AC3 and AC4.

RC2: 5. An important storm that occurred on March 17, 2013 is missing. The storm was extensively analyzed by Gkioulidou et al., 2014.

AC: The results on the RC ions during this storm (March 17, 2013) are very important for my work. They are presented in rows 7, 8, 9, 10, and 11 in Tables 1–3 (and in Fig. 1–6) and are derived from Menz et al. (2017).

The work of Gkioulidou et al. (2014), which analyzes this storm in details, is very interesting and important for understanding the dynamics of RC and it is mentioned in the several places of my manuscript. However, the results on the energy density of the RC ions obtained for this storm onboard the Van Allen Probes satellites are presented in Menz et al. (2017) more fully.

In connection with the work of Gkioulidou et al. (2014), it can be noted that hot plasma convection plays the most important role in the *formation of the general patterns in the distributions of the RC parameters during the main phase of storms*, and the work of Gkioulidou et al. (2014) focuses mainly on the analysis of a fast-variable and more localized hot plasma injection processes in the outer near-midnight regions of the RC.

Fast processes in the RC investigated in Gkioulidou et al. (2014) also appear in the figures presented in my manuscript; they are one of the main reasons for the scatter of points in these figures and the deviation of these points from general patterns (this is noted in the several places of my manuscript).

Kind regards,
Alexander Kovtyukh

---

## Author Comment (AC6)

**Table 1**

| | Satellites | $E$, keV | UT | MLT | max$\|D_{st}\|$, nT | $\|D_{st}\|$, nT | $L_m$ |
|---|---|---|---|---|---|---|---|
| 1 | Explorer-45 | 1–138 | 21.30 UT Dec 17, 1971 | 23.10 | 171 | 167 | 3.1 (Smith and Hoffman, 1973; Fig. 5, orbit 101, inbound) |
| 2 | Explorer-45 | 1–138 | 14.00 UT Feb 24, 1972 | 22 | 86 | 83 | 3.5 (Fritz et al., 1974; Fig. 2, orbit 314, inbound) |
| 3 | AMPTE/CCE | 5–315 | 15.10 UT Sept 04, 1984 | 10.30 | 64 | 46 | 4.1 (Stüdemann et al., 1986; Fig. 3) |
| 4 | AMPTE/CCE | 1–300 | 05.00 UT Sept 05, 1984 | 17.40 | 125 | 78 | 3.4 (Greenspan and Hamilton, 2002; Fig. 2a) |
| 5 | AMPTE/CCE | 30–310 | 00.20 UT Feb 09, 1986 | 17.30 | 307 | 273 | 2.8 (Hamilton et al., 1988; Fig. 7, pass 5) |
| 6 | AMPTE/CCE | 1–300 | 10.00 UT Nov 30, 1988 | 03 | 111 | 37 | 3.4 (Greenspan and Hamilton, 2000; Fig. 3a) |
| 7 | Van Allen Probes B | 10–60 | 09.56 UT Mar 17, 2013 | 19.20 | 132 | 66 | 3.2 (Menz et al., 2017; Fig. 3) |
| 8 | Van Allen Probes B | 10–570 | 10.09 UT Mar 17, 2013 | 20 | 132 | 70 | 3.6 (Menz et al., 2017; Fig. 3) |
| 9 | Van Allen Probes B | 10–60 | 18.58 UT Mar 17, 2013 | 19.30 | 132 | 98 | 3.1 (Menz et al., 2017; Fig. 3) |
| 10 | Van Allen Probes B | 10–570 | 19.00 UT Mar 17, 2013 | 19 | 132 | 98 | 3.1 (Menz et al., 2017; Fig. 3) |
| 11 | Van Allen Probes A | 10–60 | 20.08 UT Mar 17, 2013 | 19.30 | 132 | 117 | 3.0 (Menz et al., 2017; Fig. 3) |
| 12 | Van Allen Probes B | 1–300 | 07.45 UT June 1, 2013 | 01.20 | 124 | 122 | 3.0 (Kistler et al., 2016; Fig. 6) |
| 13 | Van Allen Probes B | 10–600 | 13.10 UT Aug 27, 2014 | 03 | 75 | 58 | 3.6 (Yue et al., 2018; Fig. 1) |
| 14 | Van Allen Probes B | 50–200 | 19.30 UT Mar 17, 2015 | 02 | 234 | 166 | 3.3 (Keika et al., 2018; Figs. 4, and 8) |
| 15 | Van Allen Probes B | 50–200 | 21.30 UT Mar 17, 2015 | 18 | 234 | 190 | 3.2 (Keika et al., 2018; Figs. 4, and 8) |
| 16 | Van Allen Probes A | 1–60 | 23.10 UT Mar 17, 2015 | 03 | 234 | 233 | 2.7 (Menz et al., 2019a; Fig. 1; 2019b; Fig. 6) |
| 17 | Van Allen Probes A | 10–600 | 22.10 UT Mar 6, 2016 | 05 | 99 | 98 | 3.0 (Yue et al., 2019; Fig. 1) |

---

## Author Comment (AC7)

Line 1 in Table 1

[Figure]

**Fig. 5**

(Smith and Hoffman, 1973)

Line 2 in Table 1

[Figure]

**Fig. 2**

(Fritz et al., 1974)

Line 3 in Table 1

[Figure]

**Fig. 3**

(Stüdemann et al., 1986)

Sept 5, 1984    0153-0501 UT

[Figure]

**Fig. 2**

(Greenspan and Hamilton, 2002)

[Figure]

**Fig. 7**

(Hamilton et al., 1988)

Line 6 in Table 1

[Figure]

**Fig. 3**

(Greenspan and Hamilton, 2000)

Line 7. 8, 9, 10, and 11 in Table 1

[Figure]

**Fig. 3**

(Menz et al., 2017)

[Figure]

Line 12 in Table 1

**Fig. 6**

(Kistler et al., 2016)

Line 13 in Table 1

[Figure]

**Fig. 1**

(Yue et al., 2018)

Line 14, and 15 in Table 1

[Figure]

**Fig. 4**

(Keika et al., 2018)

[Figure]

Fig. 8

(Keika et al., 2018)

Line 16 in Table 1

[Figure]

Fig. 1

(Menz et al., 2019a)

[Figure]

**Fig. 6**

(Menz et al., 2019b)

Line 17 in Table 1

[Figure]

**Fig. 1**

(Yue et al., 2019)